# Nano-palladium is a cellular catalyst for *in vivo* chemistry

Miles A. Miller[1,*], Bjorn Askevold[1,*], Hannes Mikula[1], Rainer H. Kohler[1], David Pirovich[1] & Ralph Weissleder[1,2]

Palladium catalysts have been widely adopted for organic synthesis and diverse industrial applications given their efficacy and safety, yet their biological *in vivo* use has been limited to date. Here we show that nanoencapsulated palladium is an effective means to target and treat disease through *in vivo* catalysis. Palladium nanoparticles (Pd-NPs) were created by screening different Pd compounds and then encapsulating bis[tri(2-furyl)phosphine] palladium(II) dichloride in a biocompatible poly(lactic-*co*-glycolic acid)-*b*-polyethyleneglycol platform. Using mouse models of cancer, the NPs efficiently accumulated in tumours, where the Pd-NP activated different model prodrugs. Longitudinal studies confirmed that prodrug activation by Pd-NP inhibits tumour growth, extends survival in tumour-bearing mice and mitigates toxicity compared to standard doxorubicin formulations. Thus, here we demonstrate safe and efficacious *in vivo* catalytic activity of a Pd compound in mammals.

[1] Center for Systems Biology, Massachusetts General Hospital, 185 Cambridge Street, CPZN 5206, Boston, Massachusetts 02114, USA. [2] Department of Systems Biology, Harvard Medical School, 200 Longwood Avenue, Boston, Massachusetts 02115, USA. * These authors contributed equally to this work. Correspondence and requests for materials should be addressed to R.W. (email: rweissleder@mgh.harvard.edu).

Metal coordination complexes and the subclass of synthetic organometallic compounds containing metal–carbon bonds are essential for catalysing many chemical reactions, including in biological systems. For example, prokaryotes utilize exogenous metals[1], metalloenzymes are ubiquitous in eukaryotic cells[2], copper salts are essential in catalysing bioorthogonal cycloadditions[3] and the repertoire of newer metal-enhanced ligation reactions has continued to grow. In particular, palladium (Pd) has emerged as a key transition metal used extensively in modern organic synthesis to facilitate different carbon–carbon bond formation reactions including Suzuki, Negishi, Heck, Sonogashira and Stille coupling or cross-coupling reactions. However, the implementation of this catalyst in higher organisms and live mice remains in its infancy. Recent elegant reports have used Pd compounds in bacteria[4–6], in mammalian cell culture to modify or deprotect proteins with unnatural or protected amino acids[7,8] and in prodrug-activating schemes[9–11] where delivery is relatively unproblematic. However, systemic *in vivo* usage in mammals has not been reported, likely for several reasons. First, common Pd catalysts often employ elemental metallic powders (nano-Pd cubes and associated resins)[7,10], simple salts (acetate, chloride) or phenylphosphines rather than biocompatible materials. Second, different Pd conjugates have distinct catalytic activities and systematic screening of such compounds has not been performed to identify the most potent catalysts under biological conditions. Third, the *in vivo* delivery of any such catalyst may be complicated by issues including solubility, stability and implantation in the case of resins or microparticle-based formulations. Fourth, depending on the organometallic structure, certain Pd complexes can be cytotoxic, similar to platinum complexes already in widespread use as chemotherapeutics[12], although in general the high kinetic lability of Pd compounds has limited their efficacy. Finally, delivery of nanoparticles (NPs) to cancers is often poorly understood given the limited temporal or spatial resolutions afforded by histopathology or conventional imaging, respectively. It is therefore unsurprising that biopalladium chemistry has largely been confined to cell-based systems.

We hypothesized that Pd compounds with organic ligands could be identified through screens for *in vivo* use, then be encapsulated into NP for tumoral delivery via the enhanced permeability and retention (EPR) effect[13] and once there be used to locally activate prodrugs. We systematically examined a panel of Pd compounds under biological conditions to maximize the ability to uncage allyloxycarbonyl (alloc)- and/or propargyloxycarbonyl (poc)-protected amino groups on a model chemotherapeutic prodrug (alloc-doxorubicin (DOX)) or caged fluorophores, as well as to catalyse a fluorogenic Heck coupling reaction on a model coumarin precursor (5-diethylamino-2-iodophenyl ester). We encapsulated a Pd(II) precatalyst into poly(lactic-*co*-glycolic acid)-polyethyleneglycol (PLGA-PEG) NP for *in vivo* delivery, and directly monitored the delivery and localized catalytic activity within tumours in real time via intravital confocal fluorescence microscopy. Importantly, this allowed us to temporally monitor delivery and drug activation at the single-cell level. We thus show that systemic delivery of Pd-NP in conjunction with a chemotherapeutic prodrug is feasible, results in considerable tumoral accumulation and DNA damage response, mitigates toxicity compared to standard drug formulations, and blocks tumour growth *in vivo*. These results pave the way for further studies to develop new schemes of nano-Pd chemistry *in vivo*.

## Results

### Screening finds physiologically active Pd complexes.
We first screened candidate Pd complexes likely to uncage model drugs *in vivo* in a spatiotemporally controlled manner (Fig. 1a), through reactions including deallylation, which necessarily proceeds through a Pd(0)-active species (Fig. 1b), and depropargylation, which can proceed using Pd(0) and Pd(II) forms (Fig. 1c)[8]. We tested a panel of Pd(0) and Pd(II) catalysts and precatalysts that had been reported as useful for mediating coupling reactions under relatively mild temperature, buffer and/or aerobic conditions (Supplementary Methods). Screens were performed at 37 °C using two buffers ubiquitously used in tissue culture, Hank's balanced salt solution (HBSS) and minimum essential media (MEM). Catalytic activity was monitored by a fluorogenic reaction in which either bis-alloc- or bis-poc-protected rhodamine-110 (R110; 5 μM) is uncaged via deallylation (Fig. 1d) or depropargylation, respectively, and self-immolation of the carbamate ($-CO_2$)[14,15]. Although all tested Pd compounds exhibited some degree of activity in both buffers, efficiencies were generally very low, and over 12 h, the median yield in HBSS was 2% (Fig. 1e). As a control, fluorescence measurements of R110 in the presence of Pd compounds confirmed the absence of any confounding Pd-mediated fluorescence quenching (Supplementary Fig. 1a). On average, poc cleavage was 37% less efficient compared to alloc cleavage (Fig. 1e). Regardless of alloc or poc protection, yield was on average 68% lower in the more complex MEM buffer (compared to HBSS), one of the most common media used to grow mammalian cells. In contrast, bis[tri (2-furyl)phosphine]palladium(II) dichloride ($PdCl_2(TFP)_2$) showed a 88% yield in HBSS and a 64% yield in MEM. In both cases, $PdCl_2(TFP)_2$ mediated a $>5,900$-fold fluorescence turn-on (Supplementary Fig. 1b). Dose–response characterization revealed linear relationships between $PdCl_2(TFP)_2$ and formation of fluorescent uncaged R110 product (Fig. 1f; $R^2 = 0.99$), with $>50\%$ yield across all combinations of buffers and substrate concentrations using 10 μM $PdCl_2(TFP)_2$.

The reaction of deallylation using the electron-poor phosphine ligand (TFP; tri-2-furylphosphine) to better generate the coordinatively unsaturated Pd(0) species has indeed been known[16,17] (see Supplementary Fig. 1c for scheme), but here we discover its utility in physiological conditions. TFP facilitates dissociation from the metal centre to allow available *d*-orbitals for the oxidative insertion, and promotes nucleophilic turnover of the π-allylpalladium species. Although *cis*-isomer has been isolated and crystallized using the same synthesis method used in this manuscript[18], and a 9:1 *cis–trans* isomer mixture was used herein (see Methods), many applications of TFP as a coordinating ligand utilize free TFP for *in situ* catalyst formation[16]. Fitting with this model, we found that excess TFP accelerates the reaction (Supplementary Fig. 1d). Some evidence suggests a higher affinity of TFP for *in situ* Pd(0) formation compared to triphenylphosphine[19], which may be attributed to π-back bonding, as has been observed with Ru(0) and TFP[20]. Owing to low s-donicity, TFP readily dissociates from Pd(II) species, as shown by direct equilibration experiments[21]. As one mechanism of *in situ* Pd(II) reduction, a phosphine ligand coordinated to the Pd(II) centre converts to its corresponding oxide, thereby generating the active Pd(0) species[22], and we observe TFP-oxide generation from the precatalyst under physiological conditions (Supplementary Fig. 1e). Despite observed TFP-oxide generation, $PdCl_2(TFP)_2$ was the most stable of the top performing catalysts, retaining 50% activity after 12 h preincubation in HBSS and demonstrating an average fivefold better yield than the other catalysts (Supplementary Fig. 1f).

### Nanoencapsulation enhances solubility and bioavailability.
Although $PdCl_2(TFP)_2$ is an effective reagent under physiological conditions, it is extremely lipophilic with a computed octanol–water partition coefficient ($c \log P$) of 7.0 and very poor water

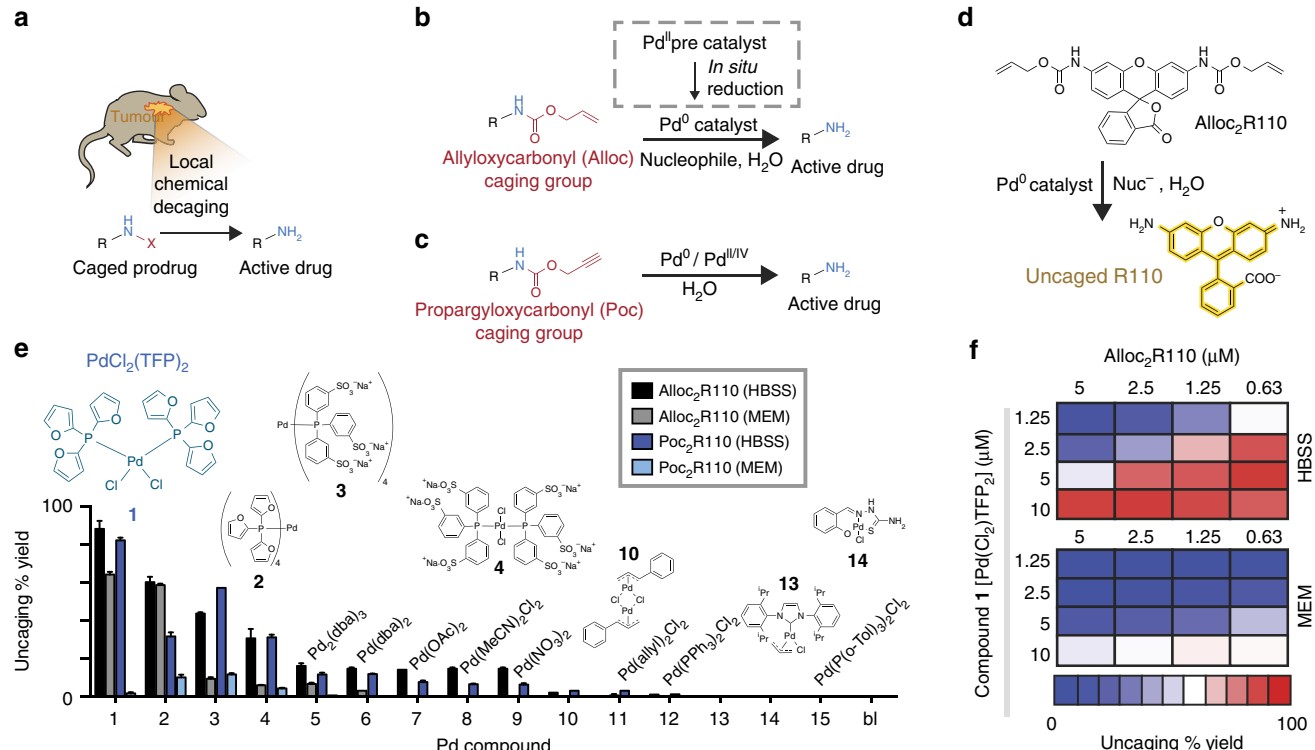

**Figure 1 | Identification of an effective Pd compound for chemistry in physiological conditions.** (**a**) Strategy of *in vivo* prodrug activation by bioorthogonal removal of a caging group X within the target tissue. (**b**) Deallylation proceeds through a Pd(0) active catalyst species, which for Pd(II) compounds can be generated *in situ*. (**c**) Depropargylation can proceed through both Pd(0) and Pd(II/IV) species. (**d**) Fluorogenic conversion of Alloc₂R110 to R110 was used as a model decaging reaction. (**e**) Pd complexes (10 μM) were screened for ability to catalyse deallylation and depropargylation via fluorogenic uncaging of Alloc₂R110 or bis(propargyloxycarbonyl) R110 (Poc₂R110; 5 μM), respectively, at 37 °C for 12 h (mean ± s.d., $n = 2$). (**f**) The most active compound **1** was tested across multiple concentrations ($n = 2$).

solubility (0.02 mg ml$^{-1}$); consequently, traditional solvent-based formulations of PdCl₂(TFP)₂ fail standard rules for 'drug-likeness' including Lipinski's Rule of Five and likely preclude its efficacy in higher organisms. Furthermore, the stability of its activity in complex physiological solutions including foetal bovine serum (FBS; Supplementary Fig. 1f), and its activity in whole tumour homogenate (Supplementary Fig. 1g), could be improved. This may in part be explained by the observation that metal complexes such as PdCl₂(TFP)₂ undergo dynamic ligand exchange processes, and ligands including TFP and Cl may be replaced by biological components (Supplementary Fig. 1c). Thus, we aimed to improve catalyst delivery while restricting its interaction with potentially reactive biological material in a clinically applicable manner. We encapsulated the precatalyst in a polymeric nanoformulation based on materials that have entered clinical trials[23] and using polymers that had previously been approved by the Food and Drug Administration for use in clinical NP preparations: PLGA and PLGA-PEG. Pd-NP was synthesized via a single-step nanoprecipitation method (Fig. 2a), forming particles 57 ± 2 nm (mean ± s.e.m., $n = 3$) in diameter as measured by dynamic light scattering (Supplementary Fig. 2a), with a polydispersity index of 0.15 ± 0.01 (mean ± s.e.m., $n = 3$) and max concentration of >30 mg ml$^{-1}$ in H₂O. Transmission electron microscopy (TEM) enabled direct visualization of Pd-NP (Fig. 2b; see Supplementary Fig. 2b,c for more images and distribution). By TEM, Pd-NP appears smaller after staining (27 ± 7 nm mean ± s.d., $n = 73$), which is consistent with similar ∼50% size decreases that have been documented for other polymeric NP[24]. Free terminal carboxylic acid on the PEG chains gave the Pd-NP a slight negative zeta-potential of −15 ± 0.9 mV (mean ± s.e.m., $n = 3$) in PBS, which has previously been reported

to improve tumoral accumulation[25,26] and mimics therapeutic micellar polymeric NPs in the clinic[27]. PLGA-PEG self-assembles to form NP with a hydrophobic PLGA core surrounded by a hydrophilic PEG outer shell[27]. The highly lipophilic PdCl₂(TFP)₂ compound loaded into the PLGA core with high efficiency (70% encapsulation; 365 nmol Pd per mg of polymer), with robust stability such that Pd-NP did not aggregate or significantly degrade over time (Supplementary Fig. 2d), and with controlled Pd release such that 50% of the compound was released from the NP by 20 h in an *in vitro* release assay (Supplementary Fig. 2e). We also measured Pd compound release across a panel of relevant solvents, solubilizing agents and biologically relevant buffer conditions. Most factors had little effect, on average changing release rates by <10% (Supplementary Fig. 2f), suggesting that the particles were stable under a variety of environments. BSA increased the release rate by 65%, likely serving as a solubilizing agent, but this effect was still modest compared to the impact of ethanol and high levels of particle-dissolving dimethylformamide (DMF) (Supplementary Fig. 2f). This evidence for NP stability and controlled release is important for successful Pd activity, since catalysis cannot occur if both the Pd compound and its substrate remain separately encapsulated in their distinct NP vehicles.

Previous reports have shown that similar polymer nano-formulations accumulate within intracellular endosomal and lysosomal vesicles, which are known to be acidic, and we confirmed that Pd-NP exhibit no significant decrease in catalytic activity at pH 5 or pH 6 ($P > 0.9$, one-tailed $t$-test, $n = 3$). Many cell types including leukocytes also exhibit mM intracellular levels of ascorbic acid[28], and we found no significant decrease in catalytic activity at 1 or 10 mM ascorbic acid ($P > 0.9$, one-tailed

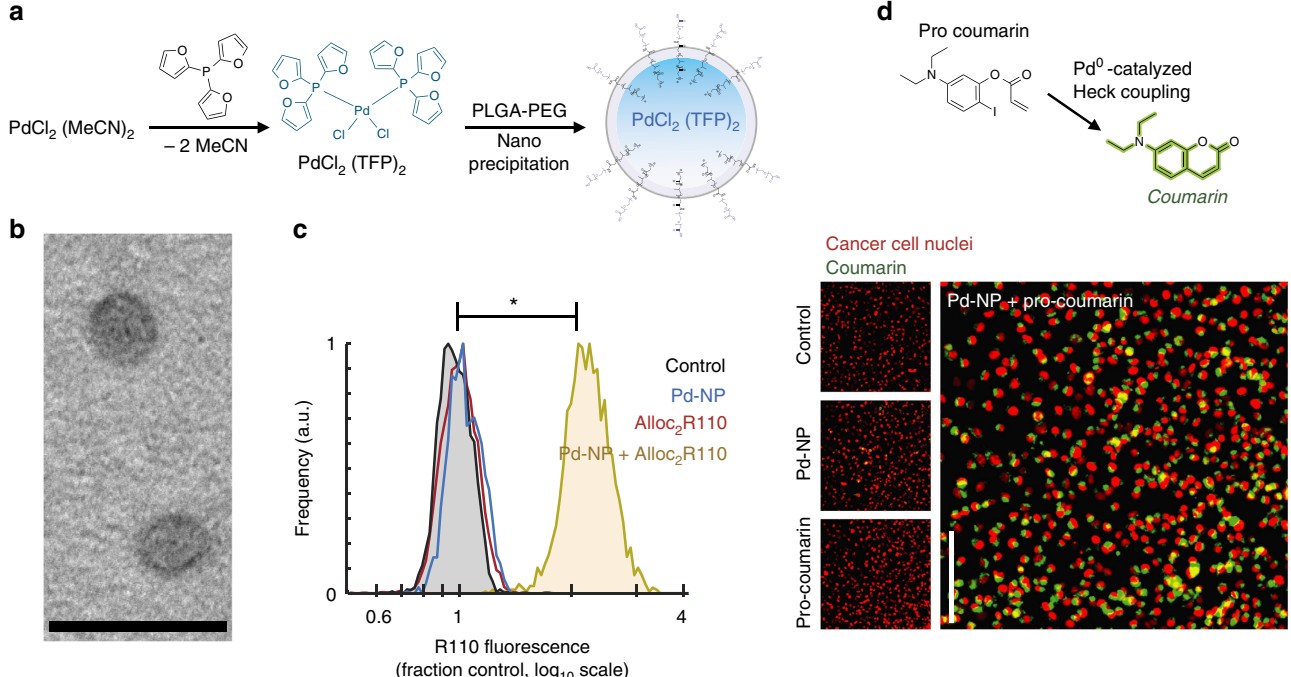

**Figure 2 | Nano-Pd mediated catalysis in live cell culture.** (**a**) Scheme for synthesis and nanoencapsulation of Pd-NP. (**b**) TEM shows spherical Pd-NP 50 nm in diameter (scale bar, 100 nm). (**c**) HT1080 cells were treated with 25 μM Pd-NP, 30 μM Alloc$_2$R110 or the combination for 24 h, and intracellular fluorescence from uncaged R110 was quantified after repeated washing by fluorescence microscopy (*$P < 10^{-4}$; two-tailed Student's $t$-test, $n = 16$). (**d**) Formation of a coumarin fluorophore (7-diethylaminocoumarin) from its non-fluorescent precursor via intramolecular Heck coupling was monitored in HT1080 cells, which were treated with 12.5 μM Pd-NP, 30 μM coumarin precursor or the combination for 24 h. Intracellular fluorescence was imaged after repeated washing, and found to be significantly brighter with the combination compared to either treatment alone ($P = 0.002$; Student's two-tailed $t$-test, $n = 18$). Scale bar, 100 μm.

$t$-test, $n = 3$). Inductively coupled plasma mass spectrometry (ICP-MS) confirmed uptake of Pd within cancer cells and showed that nanoencapsulation improved intracellular accumulation by $> 100\%$ (Supplementary Fig. 3a). Furthermore, nanoencapsulation of Pd improved the stability of its activity by $> 16$-fold after 24 h incubation in FBS, compared to unencapsulated PdCl$_2$(TFP)$_2$ (Supplementary Fig. 3b). Taken together, Pd-NP exhibit physicochemical properties similar to other PLGA-PEG-based nanomedicines being tested in the clinic[23], can be fluorescently imaged inside cells and are likely to retain activity once there.

**Nano-Pd is an effective catalyst in cell culture.** We next tested the ability of Pd-NP to activate two model fluorogenic reactions in live cell cultures. We treated cells with Pd-NP and then dosed them with a similar PLGA-PEG-based nanoformulation of bis(allyloxycarbamate) rhodamine-110 (Alloc$_2$R110) (Supplementary Fig. 3c–e), which becomes fluorescent upon Pd-catalysed allyl-carbamate cleavage (Fig. 2c). Pd-NP treatment was non-toxic, and did not significantly impact cell growth over 72 h at concentrations of up to 35 μM (concentration [Pd] throughout manuscript) across three different cancer cell lines ($P = 0.14$, two-tailed $t$-test, $n = 3$), including HT1080 and the ovarian cancer (OVCA) cell lines ID8 and A2780CP (viability = 98% ± 5%; data are means ± s.e.m., $n = 3$). The average concentration required to reduce cell count by 50% (IC$_{50}$) across these cells lines was 113 μM (Supplementary Fig. 3f), which translates to the very high dose of $> 500$ mg kg$^{-1}$ based on previous estimates of 0.5% injected dose per gram of tissue (% ID g$^{-1}$) using similar PLGA-PEG formulations and tumour models[23,29]. After 24 h incubation, HT1080 cells were washed and imaged by fluorescence microscopy, and single-cell fluorescence was

quantified by automated computational image segmentation. Results indicate clear intracellular fluorescence increase (intracellular turn-on of $> 5,000\%$) that only occurs in the presence of both Pd-NP and Alloc$_2$R110 treatment, therefore demonstrating successful activation (Fig. 2c). As a second model reaction, we again treated HT1080 with Pd-NP, but then added a precursor (5-diethylamino-2-iodophenyl ester) of a fluorescent coumarin product, 7-diethylaminocoumarin, which forms through a Pd(0)-catalysed intramolecular Heck coupling reaction (Fig. 2d)[30]. Under physiological conditions in PBS and at 37 °C, the PdCl$_2$(TFP)$_2$ precatalyst reduces to an active Pd(0) species[30], as with the alloc-cleavage reaction, which then catalyses the formation of fluorescent coumarin ($\lambda_{ex} = 375$ nm; $\lambda_{em} = 500$ nm) with rapid kinetics detectable within minutes (Supplementary Fig. 4a). In live cells, Heck coupling was measured by fluorescence microscopy after washing at 24 h after treatment (Fig. 2d), showing clear increases in intracellular coumarin fluorescence (340% fluorescence turn-on; Supplementary Fig. 4b,c) only after treatment with the combination of Pd-NP and non-fluorescent coumarin precursor.

**Nano-Pd safely accumulates in tumours.** We studied whether Pd-NP could be efficiently and safely delivered to tumours in mouse models of cancer. The maximum-tolerated dose of 50 mg kg$^{-1}$ used throughout caused no significant weight loss over several weeks of repeated administration (q4dx4; Supplementary Fig. 5a). In naive immunocompetent BALB/c mice, blood chemistry analysis showed no substantial liver or kidney toxicities at this dose, and unchanged immunoglobulin E levels suggested no allergic response to repeated (q4dx4) Pd-NP administration compared to the Pd-free PLGA-PEG NP vehicle control (Supplementary Table 1). ALP decreased slightly

(Supplementary Table 1), possibly related to artefactual haemolysis noted in some samples[31], but nevertheless remained within a normal physiological range[32,33]. Repeated Pd-NP treatment also yielded no substantial changes in complete blood counts using the fibrosarcoma xenograft model (Supplementary Table 2).

To examine how efficiently and selectively Pd could be delivered to tumours, we performed biodistribution experiments using ICP-MS. Tissue was analysed 24 h after Pd-NP administration in two xenograft mouse models. Pd-NP was intravenously injected via the tail vein in mice bearing subcutaneous human fibrosarcoma tumours. In OVCA patients, recent clinical research has highlighted the potential for intraperitoneal chemotherapy to improve outcomes[34]. We therefore administered Pd-NP intraperitoneal in an orthotopic model of disseminated OVCA using the human ES2 cell line (Supplementary Fig. 5b). In the OVCA and fibrosarcoma models, Pd accumulated in the bulk tumour mass with a % ID g$^{-1}$ of 0.50 ± 0.1 and 0.32 ± 0.04, respectively (mean ± s.e.m., $n = 3$; Supplementary Fig. 5b), corresponding to respective concentrations of roughly 7.5 and 5 µM. The OVCA disease model progresses to cause accumulation of peritoneal ascites fluid that is rich with metastatic tumour cells and associated host cells such as peritoneal macrophages, and these ascites cells collectively contained a substantial 1.25 ± 0.2% ID g$^{-1}$ of Pd (mean ± s.e.m., $n = 3$; Supplementary Fig. 5b). Thus, Pd-NP deliver catalyst to tumours with an efficiency comparable to that seen with other drug-loaded nanoformulations tested in the clinic[23], and consequently achieve tumour concentrations that are sufficient for catalysis, at least based on the in vitro studies here (Fig. 1).

In several cases, Pd-NP delivered the catalyst more selectively to the tumour compared to organs that are typical sites of toxicity for many pharmacological agents. Compared to tumour levels, Pd was substantially lower in the brain, skin, lung, bone marrow and the heart depending on the tumour model. For instance, Pd levels were eightfold higher in the tumour compared to the heart ($P = 0.002$, two-tailed $t$-test, $n = 3$) in the OVCA model; in the fibrosarcoma model, Pd levels in the bone marrow were 74% lower than those in the tumour (Supplementary Fig. 5b). These findings are important as this strategy could be used to reduce off-target organ toxicity (heart, bone marrow) limiting clinical dose escalation.

**In vivo imaging reveals nano-Pd activity in tumours.** We next examined whether Pd-NP catalysis occurred within tumours in live mouse models of cancer. We hypothesized that sequential rather than simultaneous injection of the precatalyst and its substrate could lead to more specific activity within the tumour; for instance, by limiting the ability for Pd-NP and their substrate to interact with each other at high concentrations in circulation. To test this hypothesis, we examined the biodistribution of catalytic activity using Alloc$_2$R110-encapsulated NP as a model fluorogenic substrate. In this formulation, we monitored the transport of non-fluorescent Alloc$_2$R110 by coencapsulating a near-infrared (NIR) fluorophore–polymer conjugate (PLGA-BODIPY630) that we had previously shown as effective for stably labelling NP while minimally impacting their biological activity[24].

Using the same fibrosarcoma model as described above, we found that 5 h dose-staggering of Pd-NP and their Alloc$_2$R110 substrate caused greater catalytic activity in the tumour relative to other organs, most notably the lung and heart ($P = 0.003$; total $n = 15$; pooled two-tailed $t$-test; Supplementary Fig. 6a), while not affecting relative accumulation of the Alloc$_2$R110 NP vehicle itself in these organs ($P = 0.28$; total $n = 15$; pooled two-tailed $t$-test; Supplementary Fig. 6b). In contrast, longer dose-staggering led to

a slight increase in both the accumulation and activation of the Alloc$_2$R110 NP in the liver and spleen (Supplementary Fig. 6c,d), perhaps suggesting a transient saturation in mononuclear–phagocyte system clearance in those organs by the initial Pd-NP dose. The optimal 5 h time-staggering was also tested in the orthotopic OVCA model described above, and results show substantially greater substrate activation in the tumour compared to all other organs examined except the skin (Supplementary Fig. 6e). For instance, substrate activation was 6.5-fold greater in the OVCA tumour compared to the heart. In the fibrosarcoma model, more activation was detected in the tumour compared to the heart and bone marrow (Supplementary Fig. 6e). These results surprisingly show that even though Pd-NP accumulate in the liver, kidney and spleen (Supplementary Fig. 5b), their activation of a model prodrug in these tissues is relatively low compared to the amount of activation seen in the tumour itself (Supplementary Fig. 6e). This could be due to (i) high tumour-cell uptake of both Pd-NP and their substrate, combined with (ii) metabolic inactivation of the reaction or confinement of the uptake in disparate cell populations and/or cellular compartments in clearance organs.

To more closely study catalytic activity within the tumour itself, we used time-lapse in vivo confocal fluorescence (intravital) imaging to characterize pharmacokinetics within a xenograft tumour model often used for intravital studies[35–37]. Interestingly, we found that PdCl$_2$(TFP)$_2$ causes fluorescence in polar protic solvents including aqueous solutions (max $\lambda_{ex} = 375$ nm; max $\lambda_{em} = 436$ nm; Supplementary Fig. 7a) but which is quenched by nanoencapsulation (Supplementary Fig. 7b), likely through high local concentration as has been observed in other examples[24]. Consistent with observed fluorescence, past reports have shown that halide-to-metal charge transfer transitions between Cl and Pd can participate in excitation, with phosphine ligand playing an acceptor role in metal-to-ligand charge transfer[38]. Although physical–chemical studies into photoemissive properties, especially in vivo, extend beyond the scope of this article, the fluorescence unquenching of Pd-NP is usefully visualized inside cells upon release and can thus serve as a kinetic readout (Supplementary Fig. 7c) that correlates well with dosage (Supplementary Fig. 7d; $R^2 = 0.99$).

Using a dorsal window chamber, tumours were generated via subcutaneous implantation of HT1080 cells that transgenically express a fluorescently tagged DNA damage response protein, 53BP1-mApple, which localizes to the nucleus[39] (Supplementary Fig. 8a). Owing to fluorescence quenching of the Pd as it is encapsulated within the NP, Pd-NP was not detectable in the tumour microvasculature itself. However, within minutes of intravenous injection, Pd-NP fluorescence appeared in the tumour tissue and especially in cells adjacent to the microvasculature (Fig. 3). Within tissue, intratumoral Pd fluorescence remained sustained following an initial distribution clearance away from the microvasculature (Fig. 3f; $t_{1/2\ initial} = 31 ± 4.7$ min, mean ± s.e.m. across $n = 4$), and tissue levels only decreased by an average of 10% from 1 to 1.5 h after injection ($t_{1/2\ late} > 4$ h). Consistent with the relatively slow Pd release from Pd-NP in vitro (Supplementary Fig. 2e), these data suggest that Pd is largely contained within NP while circulating in the microvasculature, but in a rapid and sustained manner Pd becomes dequenched within the tumour tissue itself. This likely occurs through cellular uptake as observed in vitro (Supplementary Fig. 7c) and as seen in similar nanoformulations, particularly in phagocytic cells within the tumour mass[40], and possibly through Pd-NP degradation within the tumour interstitium due to a combination of factors including polymer swelling, hydrolysis, mechanical stress and acidic tumour microenvironment[41].

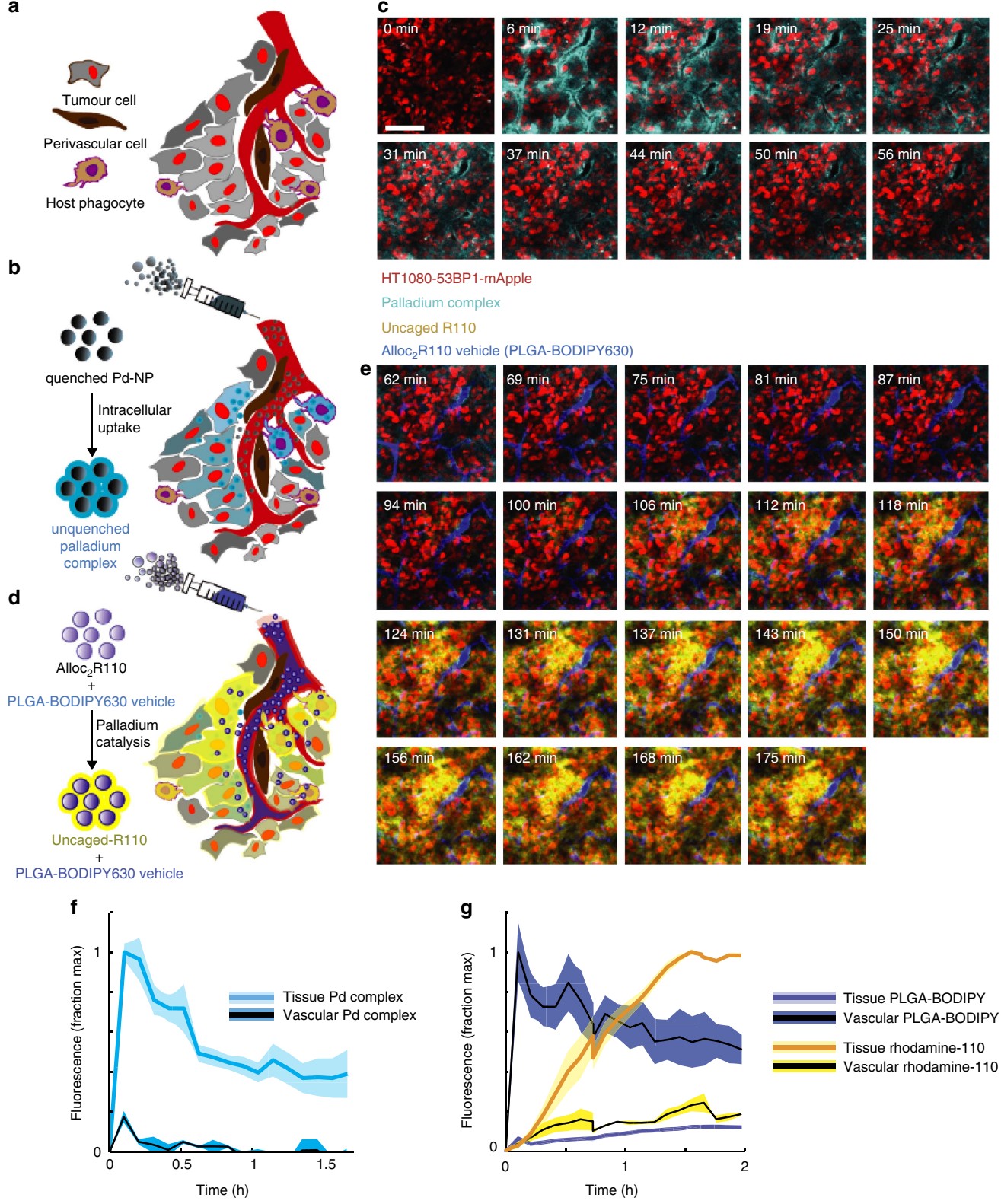

**Figure 3 | Imaging nano-Pd activity within the tumour microenvironment.** (**a,b**) Schematic of tumoral imaging strategy (**a**) before and (**b**) after intravenous administration of Pd-NP in a live xenograft model of cancer. (**c**) Representative images showing real-time fluorescence microscopy of 50 mg kg$^{-1}$ Pd-NP administration to xenograft tumours visualized through a dorsal window chamber. Postinjection elapsed time is marked. (**d**) Schematic and (**e**) representative images of intratumoral Alloc$_2$R110 delivery (25 mg kg$^{-1}$) administered 1 h following Pd-NP injection in the same tumour as in **c**. (**f,g**) Quantification of fluorescence from (**f**) Pd-NP and (**g**) Alloc$_2$R110-NP based on imaging data in **c,e**, respectively (thick lines and shading denote mean and s.e.m. across $n = 3$).

To visualize directly the activity of Pd-NP *in vivo*, we next injected NIR-labelled $Alloc_2R110$ NP as described above, 1 h following Pd-NP injection (Fig. 3e). In contrast to Pd-NP, the NIR fluorophore exhibited strong fluorescence in the micro-vasculature itself (Fig. 3g; average $t_{1/2\ vessel} = 2$ h) and gradual extravascular tumoral accumulation, such that accumulation in the tissue reached 14% peak vascular levels by 2 h after injection. Notably, the $Alloc_2R110$ NP is generated from the same PLGA-PEG polymer as used with the Pd-NP formulation, and its systemic pharmacokinetics closely resemble those of similar NP in clinical trials[23,42,43] and of clinical formulations such as PEGylated liposomal DOX (DOXIL; $t_{1/2\ initial} < 5.2$ h in humans and 2 h in rats[44,45]). Despite only a minority of $Alloc_2R110$ NP extravasating into the tumour tissue at early time points ($t < 2$ h), the majority ($68 \pm 4\%$, mean $\pm$ s.e.m. across $n = 3$) of fluorescence turn-on actually occurred within the tissue itself, and highest turn-on was observed in tumour tissue near ($< 100\,\mu m$) functionally perfused microvasculature (Supplementary Fig. 9). Either the Pd precatalyst or the $Alloc_2R110$ need to be released from their respective polymer NP vehicles in order for the catalytic reaction to occur; otherwise, the reactant and precatalyst remain isolated in their spatially separated nanoencapsulations. Single-injection control experiments showed that $Alloc_2R110$ NP

or Pd-NP alone do not show any fluorescence turn-on (Supplementary Fig. 8b–d). Both the $Alloc_2R110$ NP and Pd-NP combined are required to observe the fluorescence turn-on in the tumour tissue (Supplementary Fig. 8). In agreement with the Pd-NP imaging data that suggest Pd largely remains nanoencapsulated (and hence quenched) while circulating in the vasculature, the rhodamine imaging data demonstrate that most catalytic activity occurs locally within the tumour tissue itself (Supplementary Fig. 9). To further test the influence of nano-encapsulation on catalytic activity, we examined whether an intravenous infusion of an unencapsulated, solvent-based Pd compound formulation could similarly uncage $Alloc_2R110$ in tumours. Results show the solvent-based administration was 85% less efficient (Supplementary Fig. 8e). Additional experiments in an orthotopic and syngeneic model of OVCA extend the generalizability of Pd-NP activity, showing Pd-NP accumulation and $Alloc_2R110$ uncaging within disseminated intraperitoneal metastases in immunocompetent animals (Supplementary Fig. 8f,g).

Past reports using similar tumour models and nanoformulations have found that tumour-associated macrophages play a particularly important role in promoting vessel permeability, accumulating NP and redistributing NP payloads (for instance,

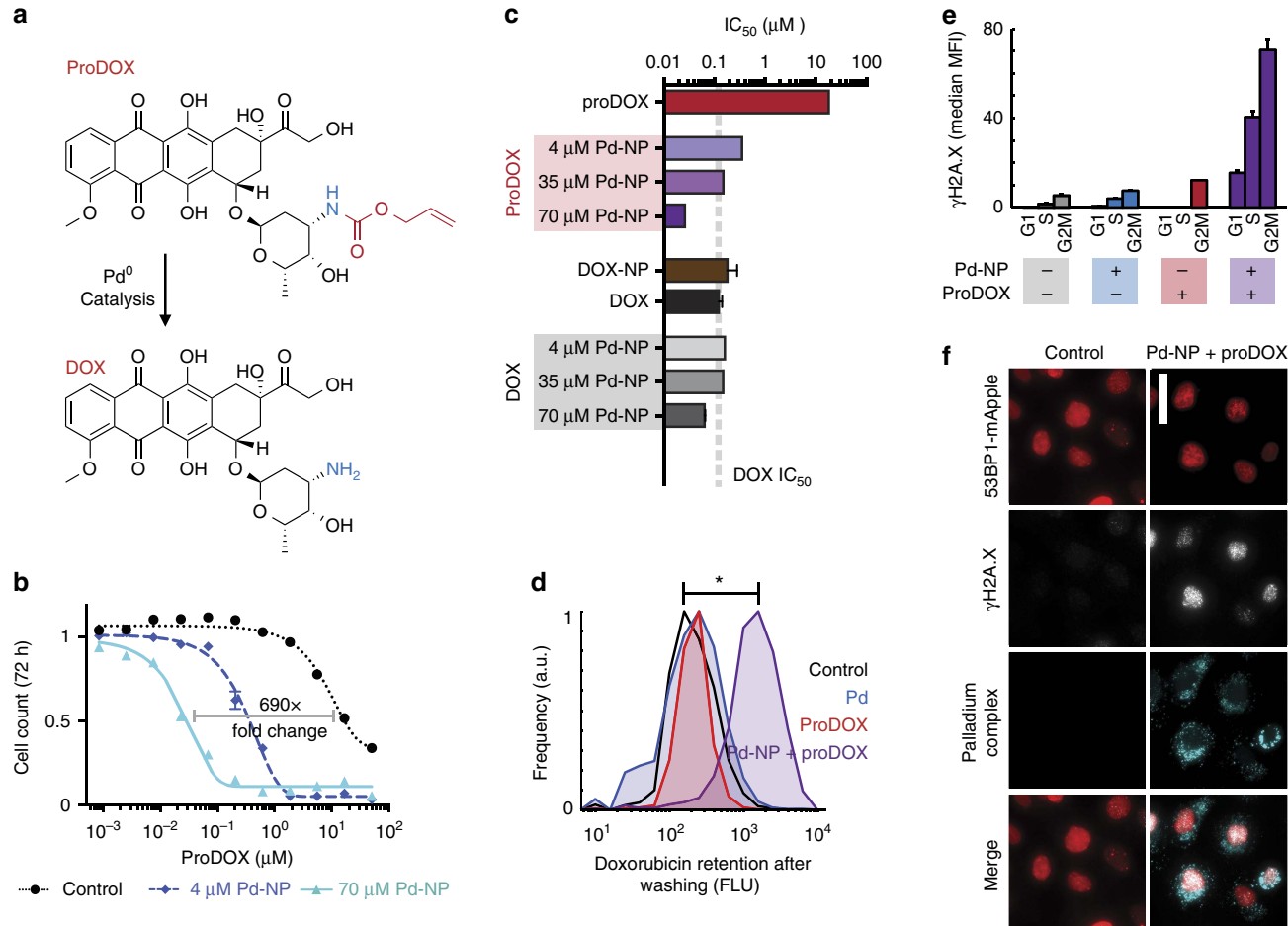

**Figure 4 | Nano-Pd activates prodrug and leads to DNA damage and cell death.** (**a**) Scheme showing the alloc-DOX prodrug (proDOX) being converted by Pd into uncaged DOX through allylcarbamate cleavage. (**b,c**) Pd-NP cotreatment enhances the concentration of proDOX causing 50% reduction in cell count ($IC_{50}$) after 72 h with HT1080 cells (mean $\pm$ s.e.m., $n \geq 2$), compared in (**c**) with $IC_{50}$ for standard solvent-based DOX or nanoencapsulated DOX-NP under the same conditions (mean $\pm$ s.e.m., $n \geq 2$). (**d**) After 24 h, 50 $\mu$M Pd-NP cotreatment with 3 $\mu$M proDOX causes the drug to be retained within HT1080 cells after washing, measured in single cells by DOX fluorescence intensity using flow cytometry ($n = 3$). (**e,f**) Fifty micromoles of Pd-NP cotreatment with 3 $\mu$M of proDOX for 24 h causes a synergistic increase in DNA damage response, as measured in single cells by (**e**) $\gamma$H2A.X staining and flow cytometry (mean $\pm$ s.e.m., $n = 3$) or (**f**) microscopy. Scale bar, 20 $\mu$m. Cell cycle was classified by DNA content.

which may include Pd catalyst or its activated products) to neighbouring tumour cells[24,40]. Here, we used intravital imaging (Supplementary Fig. 10a,b), histology (Supplementary Fig. 10c) and flow cytometry (Supplementary Fig. 10d) to find that tumour-associated perivascular phagocytes (including macrophages) indeed accumulate substantial Pd-NP and Alloc$_2$R110 NP. Imaging data suggest fluorescent product of Pd catalysis (uncaged rhodamine) accumulates in these phagocytes as well as surrounding tumour cells within several cell lengths (Supplementary Fig. 10a,b). Thus, intratumoral Pd accumulation and release via EPR effects and tumour-associated phagocyte uptake enables selective activation of its substrate locally within the tumour microenvironment and minimize systemic exposure to catalytic activity.

**Nano-Pd activates a DOX prodrug.** We next tested the ability of Pd-NP to activate a chemotherapeutic prodrug for selectively enhancing its potency. As a model drug we used DOX (Adriamycin), which is a highly efficacious front-line chemotherapeutic in wide clinical use for a broad range of cancers but has significant side effects including myelosuppression, liver damage and cardiotoxicity, a dose-limiting effect. DOX is an anthracycline that primarily works by intercalating DNA and interfering with the enzyme topoisomerase II, thus leading to DNA damage and cytotoxicity in proliferating cancer cells. The drug is covalently linked to DNA through a single bond between the 3′ amino of its sugar group (daunosamine) and the exocyclic N-2 amino of guanine[46]. Protecting this daunosamine amino group with allyloxycarbonyl (alloc) ligand (proDOX; Fig. 4a) has been shown to greatly attenuate its cytotoxicity[15]. After confirming that Pd-NP could mediate allylcarbamate cleavage on the DOX prodrug under physiological conditions *in vitro* using high-performance liquid chromatography mass spectrometry (HPLC/MS) and fluorometry (Fig. 4a and Supplementary Fig. 11a,b) and at sufficiently rapid kinetics such that >80% of prodrug can be activated within 60 min in physiological HBSS buffer (Supplementary Fig. 11c), we tested its activity in live cancer cell cultures. For improved solubility and delivery, we nanoencapsulated proDOX using the same PLGA-PEG polymers and nanoprecipitation strategy as used with the Pd and Alloc$_2$R110 NP (Supplementary Fig. 11d,f). In a cell-growth/cytotoxicity assay, Pd-NP treatment shifted the concentration of proDOX needed to reduce cell count by 50% over the course of a 72 h treatment (IC$_{50}$) by 690-fold (Fig. 4b). In HT1080 cells, Pd-NP shifted IC$_{50}$ from a relatively non-toxic 18.1 ± 0.7 μM to a highly potent IC$_{50}$ of 26 ± 1 nM (means ± s.d.; $n=2$), which is similar to potency achieved when Pd-NP was combined with the uncaged DOX, suggesting Pd-NP catalyzed nearly complete proDOX activation (Fig. 4c; Supplementary Fig. 11g). In contrast to its impact on the prodrug, 4 μM Pd-NP treatment had no effect on the potency of uncaged DOX (Fig. 4c and Supplementary Fig. 11g).

We performed flow cytometry and immunofluorescence experiments to further understand the effect of Pd-NP on proDOX activity. Because both DOX and proDOX are fluorescent ($\lambda_{ex}=488$ nm; $\lambda_{em}=560$ nm), we were able to quantify intracellular drug concentrations using flow cytometry. After 24 h treatment, proDOX was not retained in live cells after washing (Fig. 4d), indicating the prodrug had not covalently reacted with DNA. In contrast, substantial proDOX was detectable within cells after cotreatment with Pd-NP, suggesting the catalytically deprotected DOX covalently bound DNA via its 3′ daunosamine amino.

We also imaged endogenous DOX and proDOX fluorescence directly in tumour cells, and found that proDOX was preferentially colocalized with DNA only after cotreatment with Pd-NP.

This suggested that the drug was better able to bind DNA after becoming activated by the catalyst (Supplementary Fig. 12a,b). Control experiments with unencapsulated solvent-based DOX, along with DOX-NP that were synthesized to match proDOX-NP, showed similar patterns of nuclear drug accumulation. However, in contrast to the NP formulations, solvent-based DOX did not appear in perinuclear cellular compartments, which likely reflects its relative lack of uptake into endosomal–lysosomal vesicles as has been observed for similar nanoformulations[24] (Supplementary Fig. 12a,b). HPLC and fluorometry of cell lysate confirmed >50% drug activation within cells treated with both Pd-NP and proDOX by 24 h (Supplementary Fig. 12c).

As a marker of DNA double-strand break repair, we measured γH2A.X by flow cytometry (Fig. 4e) and immunofluorescence (Fig. 4f) and found that while Pd-NP and proDOX treatment alone had minimal impact, the combination caused marked increases. Cells were classified by cell-cycle phase (G1/S/G2M) according to their measured DNA content, and we found that the induced DNA damage response was most prominent in S-, G2- and M-phase cells compared to those in the G0/G1 phase, which is consistent with DOX's known cell-cycle-dependent action (Fig. 4e)[47].

**Prodrug activation blocks tumour growth and extends survival.** We investigated the ability of Pd-NP to safely activate proDOX *in vivo*, elicit enhanced tumour-cell DNA damage and slow disease progression. Using the same HT1080 xenograft tumour model as with the Alloc$_2$R110 experiments, we intravenously injected Pd-NP and waited 5 h for the Pd-NP to clear the circulation and to begin accumulating in tumour tissue. We then intravenously injected proDOX-encapsulated PLGA-PEG NP that had been labelled with an NIR fluorophore to track their delivery (native proDOX fluorescence was insufficiently bright to directly image in these *in vivo* experiments). By 24 h after treatment, tumours displayed detectable accumulation of Pd-NP, proDOX NP and elevated DNA double-strand-break response as measured by focal accumulation of the 53BP1-mApple within nuclei, which generally indicates DNA damage response at a site of non-homologous end-joining repair (Fig. 5a,b). In contrast to the combination treatment, either Pd-NP or proDOX NP alone had little impact on DNA damage response (Fig. 5b). To measure effects on longitudinal disease progression, we also measured tumour size daily over the course of 8 days following NP treatment and found no significant effect of either Pd-NP ($P=0.45$) or proDOX NP treatment ($P=0.98$) alone ($n\geq15$; two-tailed Mann–Whitney U-test for both), but the combination treatment resulted in slower tumour growth than control and single-treatment groups (Fig. 5c; $P=0.011$). Given the favourable safety profile of the Pd-NP and proDOX treatment, we increased proDOX dose to 300% the maximum-tolerated dose (MTD) of unencapsulated solvent-based DOX in nu/nu mice, from 16 μmol kg$^{-1}$ (which equates to 10 mg kg$^{-1}$ DOX[48]) to 48 μmol kg$^{-1}$. Of note, even the lower dose of 38.5 μmol kg$^{-1}$ liposomal DOX NP (DOXIL) has been found to cause severe and prohibitive toxicity in 100% of animals treated in similar mouse models[49]. In contrast, Pd-NP combined with the 48 μmol kg$^{-1}$ dose of proDOX showed efficacy that was equivalent to treatment with either or nanoencapsulated or solvent-based DOX (Fig. 5c). These results were further substantiated by tumour-weight measurements at the end of treatment (Fig. 5d and see Supplementary Fig. 13a for representative tumour photos). Using an orthotopic model of disseminated OVCA, we confirmed that the combination of Pd-NP and proDOX treatment could extend survival in tumour-bearing mice (Supplementary Fig. 13b) and reduced ascites formation as

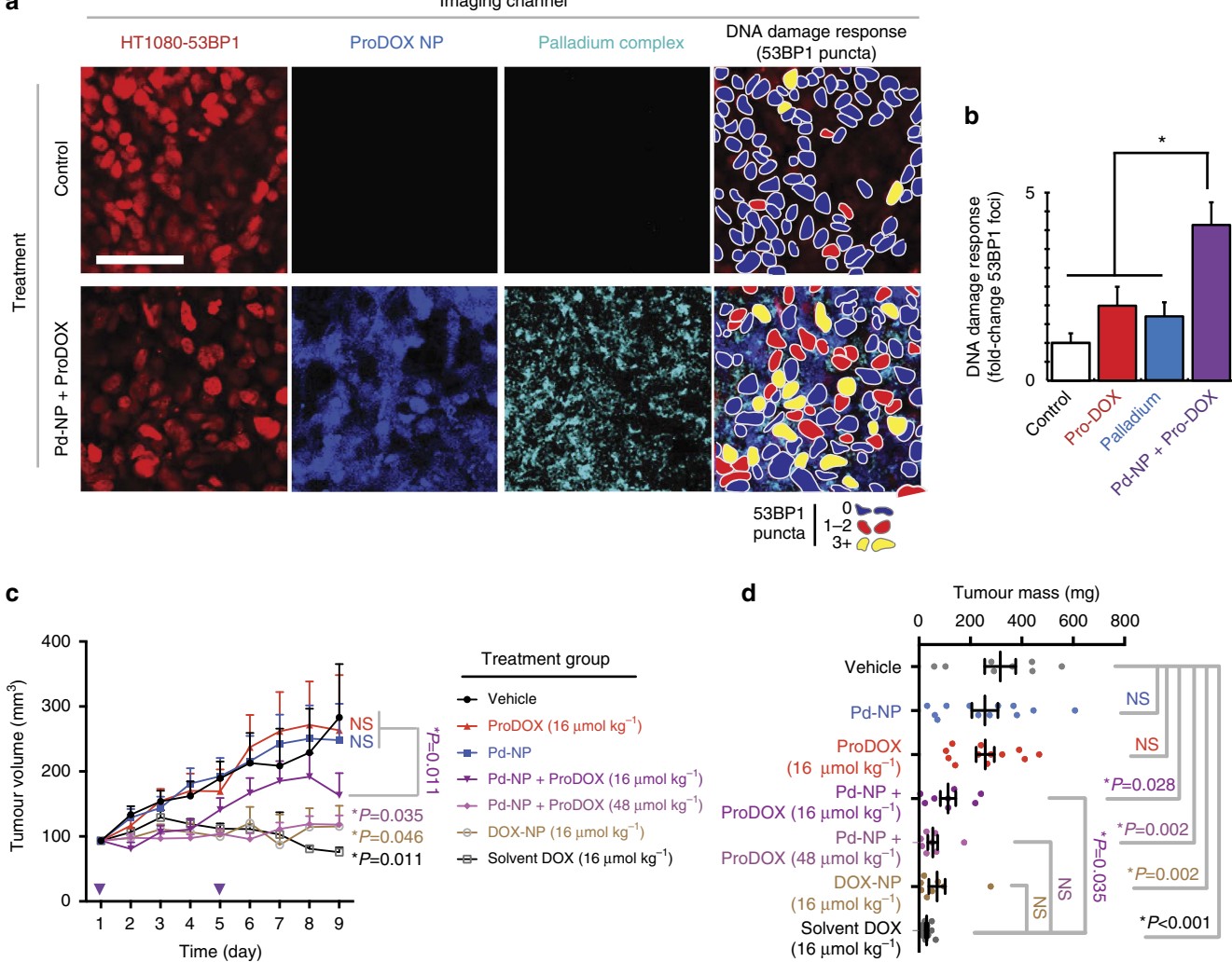

**Figure 5 | ProDOX activation causes DNA damage *in vivo* and blocks tumour growth. (a)** Fluorescence microscopy of tumours 24 h following treatment with 50 mg kg$^{-1}$ Pd-NP, followed by 16 µmol kg$^{-1}$ (~10 mg kg$^{-1}$) treatment with fluorescently-labelled proDOX NP 5 h later. At right, cells were pseudocoloured based on visible 53BP1 puncta, which indicates DNA damage response. Scale bar, 50 µm. **(b)** Combined Pd-NP and proDOX treatment elicits increased DNA damage response at 24 h after treatment in HT1080 tumours (mean ± s.e.m.; *$P = 2 \times 10^{-5}$; two-tailed *t*-test; $n = 4$). **(c,d)** Combined 50 mg kg$^{-1}$ Pd-NP and proDOX treatment blocks tumour growth in HT1080 tumours, measured by **(c)** a calipre, **(d)** final weight of excised tumours and representative images (Supplementary Fig. 13a). Purple arrows mark treatment days. For **c,d**, data are means ± s.e.m. (*two-tailed Mann–Whitney *U*-test, $n \geq 8$; NS = not significant).

efficiently as treatment with the parent drug DOX at its MTD (Supplementary Fig. 13c).

Quantification of proDOX activation within both OVCA and fibrosarcoma tumours was performed using HPLC, and revealed an average intratumoral prodrug activation of 48 ± 15% of the drug depending on tumour model (Supplementary Fig. 13d), thereby confirming roughly equivalent efficacies between 16 µmol kg$^{-1}$ DOX and the combination of Pd-NP with 48 µmol kg$^{-1}$ proDOX. To more directly visualize prodrug activation, we used confocal microscopy to image drug fluorescence within tumour nuclei. We treated an orthotopic OVCA xenograft model with either solvent-based DOX, DOX-NP, proDOX or the combination of Pd-NP and proDOX. At 24 h after treatment, we excised tumours, formalin-fixed and cleared tissue for fluorescence microscopy, counterstained with Hoechst 33342 to visualize DNA and imaged localization of DOX or its prodrug to quantify colocalization with DNA (Supplementary Fig. 13e,f). Treatment with proDOX alone did not lead to significant nuclear drug localization and instead accumulated in the cytoplasm, suggesting that the prodrug did not covalently

bind DNA and remained alloc-protected *in vivo*. In contrast, with Pd-NP cotreatment, proDOX exhibited similar nuclear accumulation as seen with DOX and DOX-NP, thus confirming drug activation.

**Nano-Pd locally and safely activates prodrug in the tumour.** We hypothesized that Pd-NP safely achieved efficacy by accumulating within tumours and activating prodrug locally, while minimizing systemic exposure of the activated drug elsewhere in the body. As with many chemotherapeutics, DOX is known to cause haematological toxicities. Therefore, we examined circulating lymphocyte and reticulocyte counts from whole blood in fibrosarcoma-bearing mice following two courses of treatment (q4dx2). Only unencapsulated, solvent-based DOX caused significant lymphocytopenia, while both solvent-based DOX and DOX-NP led to total reticulocyte ablation, which indicates bone marrow toxicity (Fig. 6a). Similarly, previous reports have found indications of haematological toxicity with 20–38.5 µmol kg$^{-1}$ liposomal DOX NP (DOXIL) in similar

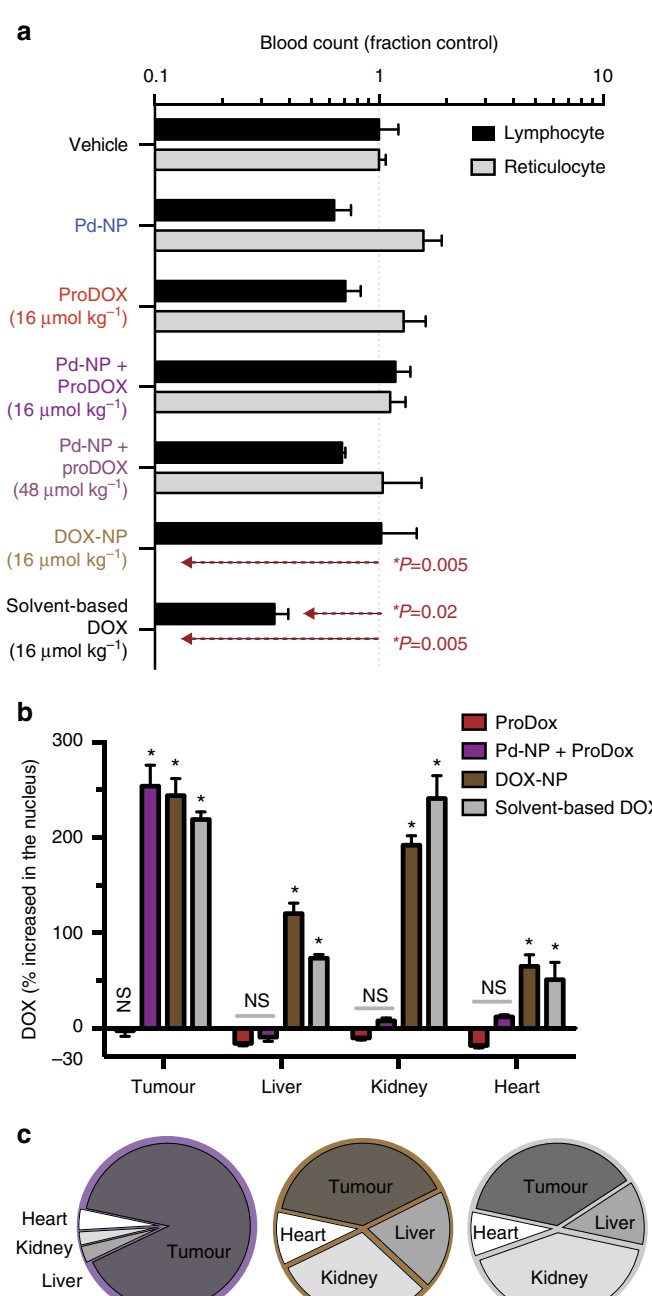

**Figure 6 | Nano-Pd safely and selectively activates proDOX in the tumour.** (**a**) Lymphocyte and reticulocyte counts from whole blood in nu/nu mice bearing HT1080 tumours after treatment (q4dx2) show haematological toxicity with standard solvent-DOX and DOX-NP formulations (*$n = 3$, two-tailed $t$-test, mean ± s.e.m.). (**b**) Pretreatment with Pd-NP followed by proDOX leads to drug accumulation in tumour nuclei of orthotopic A2780CP intrabursal ovarian tumours, but not in the nuclei of key organs related to toxicity. At 24 h after treatment, tumours and organs were excised and imaged for colocalization between intrinsic DOX fluorescence and DNA (via Hoechst 33342 counterstain; mean ± s.e.m., $n \geq 4$; *$P < 0.05$, two-tailed $t$-test; NS, not significant; see Supplementary Figs 11 and 12 for images). (**c**) Pie chart showing distribution of nuclear DOX accumulation in orthotopic ovarian tumours and key organs related to toxicity, corresponding to (**b**) ($n \geq 4$).

mouse models[49]. Our toxicity data are supported by HPLC analysis of excised tissue, showing that the prodrug activation strategy leads to <10% the levels of activated DOX in bone marrow compared to solvent-based DOX or DOX-NP treatments (Supplementary Fig. 14a). Although proDOX is detectable in the bone marrow (Supplementary Fig. 14b), little activation is detectable (Supplementary Fig. 14a), likely owing to the aforementioned results showing Pd accumulation is relatively low in the bone marrow (Supplementary Fig. 5b). As further evidence, no detectable activation of Alloc$_2$R110 NP was observed in the bone marrow (Supplementary Fig. 6e). In other organs such as the heart, nanoformulation of DOX somewhat mitigates accumulation but not nearly as much as the Pd-NP activation strategy (Supplementary Fig. 14a).

As with the tumour analysis above, we used fluorescence microscopy to investigate the degree to which Pd-activated proDOX accumulated in cell nuclei of potential organs of toxicity. Using the same model and protocol as used in the tumour analysis (Supplementary Fig. 13e,f), we found that proDOX alone did not lead to significant nuclear drug localization in any of the examined organs and instead accumulated in the cytoplasm, suggesting that the prodrug did not covalently bind DNA *in vivo* (Fig. 6b,c and Supplementary Fig. 14). HPLC analysis of excised tissue confirmed the absence of detectable proDOX activation in the skin, bone marrow, heart, brain (Supplementary Fig. 14a) and tumour (Supplementary Fig. 13d) when Pd-NP was not coinjected. However, in other organs including the liver and kidney, some activated DOX was observable, likely owing to drug metabolism (Supplementary Fig. 14a), but this did not lead to nuclear drug accumulation as detectable by microscopy (Supplementary Fig. 14). In contrast to proDOX, treatment with fully activated solvent-based DOX and DOX-NP led to strong and selective nuclear accumulation in the tumour and all other organs, including the liver and heart where there has been recognized drug toxicity issues (Fig. 6b,c). Encouragingly, combined treatment with Pd-NP followed by proDOX led to significant nuclear drug accumulation in the tumour but not in other organs (Fig. 6b,c and Supplementary Fig. 14). We imaged Pd compound accumulation and activity in the liver and kidney to better understand why nuclear drug accumulation was not observed in these organs. We found that although Pd-NP and its substrate indeed accumulate, in the liver this uptake is primarily confined to resident phagocyte populations rather than hepatocytes (Supplementary Fig. 15). Furthermore, imaging suggests that the Pd compound and its substrates may accumulate in different cellular or subcellular compartments in both the kidney and the liver (Supplementary Fig. 15), which could potentially be the result of active drug pumps highly expressed in those tissues. This is in contrast to what is observed in tumour cells, where both the Pd compound and substrate fluorescence show substantial spatial overlap (Supplementary Figs 8–10). No significant body weight was lost in all treatment groups, even when using the proDOX at 300% the equivalent MTD of solvent-based DOX (Supplementary Fig. 14e). Overall, these data provide evidence that (i) Pd-NP activate the prodrug within the tumour, thereby allowing formation of covalent DOX–DNA adducts; and (ii) enhanced NP accumulation and local catalysis via EPR effects in the tumour effectively mitigate exposure to active drug elsewhere in the body, particularly the heart and bone marrow.

## Discussion

Our results show that nanoencapsulated Pd can be delivered to tumours and mediate chemical reactions within the local tumour microenvironment. Once there, Pd can catalyse the conversion of innocuous prodrugs to more cytotoxic metabolites in a time and

locally constrained manner, thereby minimizing systemic exposure and toxicity. Specifically we show that nanoencapsulation enhances $PdCl_2(TFP)_2$ bioavailability by increasing its delivery to tumour cells and tumour-associated phagocytes; Pd-NP can perform diverse fluorogenic and prodrug-activating reactions in live-cell cultures, including allylcarbamate cleavage and Heck cross-coupling reactions; Pd-NP activate proDOX *in vivo*, allowing it to stably bind and damage DNA in tumour cells, and this occurs locally within the tumour; and *in vivo* Pd-NP catalysis is an effective and safe treatment that halts tumour growth, extends survival and limits bone marrow toxicity in orthotopic tumour models. Our studies shed light on some key pharmacological properties and principles as *in vivo* Pd chemistry is expanded to other applications. In particular, high-resolution intravital microscopy revealed rapid tumoral accumulation of Pd-NP and demonstrated their activity within minutes of substrate addition. Surprisingly, catalysis occurred not within or immediately adjacent to tumour microvasculature (for instance, in endothelial cells), but rather in a more diffuse manner within several cell lengths of the vessels. These findings highlight the importance of the EPR effects in governing localized Pd-NP activity, the majority of which occurs after the NP have extravasated into the tumour microenvironment. Once extravasated, nanoencapsulated precatalyst and substrate accumulated in both tumour cells themselves and neighbouring perivascular phagocytes, including tumour-associated macrophages that have been observed to contribute substantially to tumoral NP accumulation and from which therapeutic NP payload can locally diffuse[24,40]. Overall, these results have implications for understanding how EPR effects drive Pd-NP catalysis on a local level within tumour tissue, and highlights the potential of future immunological applications for targeting resident phagocyte populations.

As a member of the platinum group metals, salts of Pd(II) can exert cytotoxic effects via DNA crosslinking[12], just as various platinum[50], ruthenium[51], rhenium[52] and osmium[53] compounds have been developed as anticancer cytotoxics. The direct antiproliferative effects of Pd are dose and compound specific, and $IC_{50}$ in cell culture range from $>100\,\mu M$ to 10 nM for the most effective compounds[12]. Compared to the highly cytotoxic Pd compounds, the nanoencapsulated tri(2-furyl)phosphine-based catalyst used in our studies exhibited a relatively low $IC_{50}$ of $70-170\,\mu M$ depending on the cell line. Based on ICP-MS observation of $0.3-1.25\%$ $ID\,g^{-1}$ in tumour cells of multiple xenograft models (Supplementary Fig. 5b), we found an intratumoral Pd-NP concentration of $5-20\,\mu M$ at the MTD of $50\,mg\,kg^{-1}$ (calculated from $\%\,ID\,g^{-1}$ and assuming standard tissue density), which is well above the concentrations at which we observed efficient catalytic activity, yet no organ accumulated Pd-NP at levels observed to be toxic *in vitro*, and no toxicity was observed in treated animals. It is this safety window that potentially allows the use of Pd chemistry in other applications. As some precedent for safe clinical use, a Pd(II) complex with a strongly chelated organic ligand has reached phase-I/II clinical trials for application to photodynamic therapy[54]. In this work, we show that Pd-NP accumulate somewhat selectively within tumours (Supplementary Fig. 5b), and thus it is relevant that radioactive $Pd^{103}$ seeds and Co-Pd thermal seeds have been used for local irradiation and hyperthermia, respectively, in phase-I/II trials for various solid tumours and have compared favourably in terms of safety and feasibility[55].

*In vivo* Pd chemistry represents an exciting opportunity to expand on evolving bioorthogonal and other reactions. We envision the expanded use of Pd chemistry in prokaryotic synthetic biology and metabolic engineering[56], protein engineering[57], as well as for other medical and engineering applications in eukaryotes beyond cancer. For example, it is conceivable to fabricate Pd implants to catalyse local reactions in wound healing, antiproliferative and bacteriostatic applications; to deposit Pd gels in postsurgical sites; or to locally deliver such materials via catheter-based approaches. Here we show that nanoencapsulated Pd can be safely and efficiently administered, and further optimization could be performed to treat specific cellular populations or tissues through molecular NP targeting[58]; to combine Pd-NP with alternative and complimentary substrate delivery methods; to coencapsulate additional drugs or adjuvants for combination therapy and enhanced EPR effects[59]; or to directly attach Pd catalyst to adoptively transferred immune cells[60]. Irrespective of the exact application, we foresee a number of exciting opportunities in expanding *in vivo* Pd chemistry.

## Methods

Additional Materials and methods can be found in the Supplementary Materials, including nuclear magnetic resonance (NMR) analyses (Supplementary Figs 16–29).

**Cell lines and animal models.** All animal research was performed in accordance with guidelines from the Institutional Subcommittee on Research Animal Care. Appropriate experimental sample sizes were estimated from previous imaging and longitudinal tumour growth data in similar animal models[24,37,40,61]. All implantations were performed in 50 μl PBS using 6–8-week-old female mice. For HT1080 xenografts, two million cells were subcutaneously injected on flanks of nu/nu mice (Cox7/MGH), and tumour size was measured by four calipre measurements per tumour from two blinded researchers. Animals were ranked by tumour size and evenly assigned to treatment groups. Drug treatment began roughly 3 weeks postimplantation once tumours reached an average diameter of $5.1 \pm 1.7\,mm$ (mean ± s.d.; $n = 110$). Animals not receiving Pd-NP or proDOX/DOX NP received drug-free PLGA-PEG NP vehicle controls. NP formulations were made fresh before each injection, and batches were tested for size and monodispersity by dynamic light scattering, TEM and appropriate drug release as described previously. According to preestablished criteria, mice were killed when tumour burden reached over 1 cm in diameter, or 2 cm in diameter if only one tumour was present, or according to a body condition score of 2. For orthotopic OVCA imaging, $10^5$ cancer cells were intrabursally injected and intraperitoneally drug treatment began roughly 7 weeks postimplantation once tumours became palpable (with A2780CP tumours in nu/nu mice) or ascites formed (with ID8 tumours in C57Bl/6 mice). As a model of disseminated intraperitoneal disease, 10 million ES2 OVCA cells were intraperitoneally injected in nu/nu mice. For assessing survival in ES2 xenografts, animals were injected 3 days postinoculation with $50\,mg\,kg^{-1}$ Pd-NP followed by $20\,mg\,kg^{-1}$ proDOX 2 h later, with abdominal swelling and/or body condition score of 2 as the experimental end point. For assessing ascites at 14 days postimplantation, animals were treated on day 3 with $10\,mg\,kg^{-1}$ soluble DOX or the combination of $50\,mg\,kg^{-1}$ Pd-NP followed by $30\,mg\,kg^{-1}$ proDOX 2 h later. Biodistribution (HPLC; ICP-MS) was likewise assessed at 14 days postimplantation, 24 h following drug treatment in the ES2 model. Blood chemistry toxicity was performed in naive 6–8-week-old female BALB/c mice. Drug-induced weight loss did not exceed 10% in any treatment group for any model. Cell lines were obtained directly from ATCC (HT1080; ES2), Sigma (A2780CP) and through generous provision from Dr Katherine Roby (University of Kansas) for ID8 (ref. 62), were routinely cultured according to the provider's guidelines and were not independently verified. Transgenic cell lines were generated from lentiviral infection and puromycin selection as described previously[61]. All cell lines underwent mycoplasma and mouse antibody production testing (Bioreliance Corp.). For all procedures, mice were anesthetized with an isoflurane vapouriser on a heated stage; killing was performed by $CO_2$ chamber when necessary, and all treatment groups underwent procedures and monitoring consecutively on the same day when possible, but in a randomized order.

**Intravital microscopic imaging.** In HT1080 xenograft experiments, NP was injected via tail-vein catheter immediately after mixing to a final $1 \times$ PBS solution. Intravital microscopy was performed on an Olympus FV1000 Confocal-Multiphoton Imaging System using a XLUMPLFLN $\times 20$ water-immersion objective (NA 1.0; Olympus America); $2 \times$ digital zoom; sequential scanning using 405-nm, 473-nm, 559-nm and 635-nm diode lasers and a DM405/473/559/635-nm dichroic beam splitter; and collection of emitted light using beam splitters (SDM473, SDM560 and/or SDM 640) and emission filters BA430-455, BA490-540, BA575-620 and BA655-755 (all Olympus America). Dorsal window chamber imaging was performed following previously described procedures[61], such that two million HT1080-53BP1-mApple cells were suspended in 50 μl PBS, injected under the fascia of nu/nu mice (Cox7; MGH) 30 min after surgical chamber implantation and imaged 2 weeks later. Terminal orthotopic imaging was performed by

surgically opening the peritoneal cavity 24 h following intraperitoneal drug injection.

**Nanoformulation.** Pd-NP was synthesized in a single-step nanoprecipitation method by first combining 12 mg $PdCl_2(TFP)_2$, 30 mg PLGA(75:25 lactide: glycolide)$_{8.3\,kDa}$-PEG$_{5.5kDa}$ (Advanced Polymer Materials Inc.) and 6 mg PLGA(50:50 lactide:glycolide)$_{30-60\,kDa}$ (Sigma) in a 3 ml mixture of 1:1 DMF:acetonitrile, which was then added dropwise to 90 ml $H_2O$ under stirring at room temperature for 4 h. Aqueous solution was sequentially filtered through 40 μm nylon mesh strainer (Falcon Corning) and then through a 0.45 μm cellulose acetate syringe filter (Cole-Parmer), and concentrated in Amicon 100 kDa molecular-weight-cutoff centrifugal filters (Millipore) spun at 3,000 $g$ for 30 min. ProDOX, DOX and Alloc$_2$R110 NP were formulated using 9 mg drug, 100 mg PLGA-PEG and 20 mg PLGA dissolved in a 10 ml 1:1 DMF:acetonitrile solution, which was then added to 250 ml $H_2O$ and filtered as with Pd-NP. In cases where intravital imaging was used, PLGA-BODIPY630 was used instead of PLGA (described previously[24,40]). Concentrated NP was washed two times with deionized $H_2O$ and resuspended in 1 ml of nuclease-free $H_2O$. NP was diluted 20 × in water or PBS and subjected to size and zeta-potential measurements using dynamic light scattering (Malvern Zetasizer); zeta-potentials in dI-$H_2O$ ( $-16 \pm 0.2$ mV, mean $\pm$ s.e.m., $n = 3$) and PBS ( $-15 \pm 0.9$ mV, mean $\pm$ s.e.m., $n = 3$) were not significantly different ($P = 0.3$, two-tailed $t$-test). NP drug or precatalyst loading was determined by absorbance (Nanodrop spectophotometer) and interpolation from a standard curve ($R^2 > 0.99$) after 1:10 dilution in DMF. Drug or precatalyst loading efficiency was defined here as the fraction of initial drug used in the nanoprecipitation reaction that was successfully encapsulated and recovered in final NP product.

TEM experiments were performed on a JEOL 1011 electron microscope, and sample was prepared by depositing 10 μl of Pd-NP (1.0 mg ml$^{-1}$) onto a carbon-coated copper grid. Excess solution was blotted and grids were stained with phosphotungstic acid, which was then blotted, dried and immediately imaged.

**Data availability.** The data that support the findings reported herein are available on reasonable request from the corresponding author.

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

## Acknowledgements

This work was supported in part by the US National Institutes of Health (NIH) Grants RO1-CA164448, P50-CA086355, PO1-CA139980, U54-CA151884 and K99-CA207744. B.A. was supported by a grant from the Deutsche Akademie der Naturforscher, Leopoldina. We thank Dr P. Caravan (MGH) and O. Pinkhasov (MGH) for ICP-MS assistance, and Dr J. Carlson for discussion and assistance.

## Author contributions

M.A.M., B.A., H.M. and R.W. developed the concepts and designed the experiments; M.A.M., B.A. and H.M. synthesized, characterized and tested the Pd-NP and substrates; M.A.M., D.P. and R.H.K. performed imaging experiments; M.A.M. performed the *in vivo* safety and efficacy studies; M.A.M. generated the cell lines; M.A.M. and R.W. wrote the paper; all authors analysed results, wrote the Methods section and edited the manuscript.

## Additional information

**Competing interests:** The authors declare no competing financial interests.

