## [Peer Review File · Nature Communications]

Reviewers' comments:

Reviewer #1 (Remarks to the Author):

The manuscript of Weissleder describes a novel concept, namely a nanoformulation of palladium that targets a tumor and subsequently catalyzes the conversion of a prodrug in a drug which leads to tumor regression. This is certainly an interesting approach and the authors have investigated in depth the performance of the formulation. The question is however whether this an approach potentially suitable for clinical translation. The following questions are raised and remarks are made.

1. The palladium catalyst was loaded in pegylated PLGA nanoparticles. It is know that from many studies that such particles accumulate in tumor by the so-called EPR effect. However at most 1 % of the injected dose (particularly in humans) end up in the tumor, This means that the particles are also deposited in other tissues and organs. The Pd-loaded nanoparticles will cause toxic side effects after administration of the prodrugs.

2. It is not clear to what extent this approach results in better therapeutic outcome as compared to nanoparticles loaded with a cytostatic drug. Figure 5c shows that a nanoparticle formulation of PLGA-PEG of and as well as free drug are as effective in inhibition of the tumor growth.

I therefore do not see therapeutic advantages (better efficacy and/or reduced toxicity compared to standard chemotherapies).

Reviewer #2 (Remarks to the Author):

In the manuscript "Nano-palladium is a cellular catalyst for in vivo chemistry", Miller et al. presented a novel strategy to in situ activate the prodrug using palladium catalyst based chemistry. This work is of interest and novel in nanomedicine. Overall, the authors did very careful studies in terms of chemistry and biology. However, I have several concerns. I would like to recommend major revision.

Comments:

1. What is the loading of catalyst in PLGA-PEG NPs and their stability?
2. It seems that buffers can significantly impact the activity of catalyst. The authors screened the catalyst using HBSS and MEM. Why do not use tissue homogenates?
3. It is very surprising that the activation of prodrug only occurred in tumor while not in other organs, such as liver and spleen. The mechanism must be clarified clearly.
4. Why were NPs administered into OVCA tumor intraperitoneally and subcutaneous human fibrosarcoma tumors intravenously? The authors should discuss why NPs were administered differently.

Reviewer #3 (Remarks to the Author):

The manuscript reports on the catalytic prodrug activation that is triggered by a palladium complex, which was incorporated in nanoparticles. On the one hand the general impression is that the work shows significant problems regarding chemistry and structural characterization. On the other hand the manuscript reports on a very impressive biological study, which sheds light on the biological properties of palladium containing nanoparticles and their catalytic properties in-vitro and in-vivo. Despite the fascinating biological results, the work does not stand on a very solid basis as the most important compound(1) was not characterized in an appropriate manner. Therefore, recommendation for publication can not be given.

The critical remarks in more detail:

- 1) The purity of the palladium complexes was not determined. This a "no go" regarding medicinal chemistry and drug discovery. Even small impurities can influence the outcome of biological tests significantly. Especially when dealing with probably non stable coordination compounds, high purity of all test compounds is critical.
- 2) The structural characterization of the new compounds lacks sufficient evidence.
 - a) For example, in the ¹H NMR data of compound 1 several signals with an intensity of 0.3 were observed. What is the origin of these signals ?
 - b) There are no mass spectrometry data and there is no experimental evidence for the presence of the chlorine ligands of 1.
 - c) The aspect of cis/trans isomerism was not evaluated appropriately. Figure 2a shows a trans-configured synthesis precursor and a cis-configured complex PdCl₂(TFP)₂. P-NMR data of 1 suggest a mixture of cis- and trans-isomers with the trans-isomer as impurity.
 - d) The structure of compound proDOX was only checked by NMR spectroscopy, which is insufficient as a stand-alone characterization technique.
- 3) The aspect of stability and metabolism under biological conditions was not touched at all. Metal complexes can undergo ligand exchange processes in biological systems. The Cl-ligands of 1 might be easily replaced by components of the cells or tissues resulting in different Pd-containing species.
- 4) In the discussion section a 5-20 micromolar intratumoral Pd-NP concentration is mentioned. How was this calculated ?

Reviewer #4 (Remarks to the Author):

The manuscript by Miller et al describes palladium (Pd) complex formulated with polymer nanoparticles (NPs) as catalyst for chemical reactions in vivo. The authors claim that the obtained nano-palladium is able to catalyze activation of fluorescent dyes and liberation of active drug at the cellular and animal levels. It was also shown that nano-palladium in combination with doxorubicin prodrug can significantly improve specific targeting of tumors in mice: the active drug was liberated specifically inside tumors producing inhibition of the tumor growth with minimized toxicity in other organs, in contrast to conventional formulations of this drug. The work presents several novel claims. It was shown that nano-palladium can operate in physiological conditions and even more importantly, it can catalyze reactions in mice, which is probably the key novelty of this work. Moreover, the important novel observation is the remarkable specificity of tumor targeting with minimal toxicity in other tissues, which is connected to the drug activation through Pd-catalysis. This work will be definitely of interest for a broad community of researchers. It will be of interest for chemists working on metal-catalysis and those working on bio-orthogonal reactions. It will be also of interest for biologists, nanotechnologists, pharmacologists and biomedical researchers, developing new tools against cancer, etc. The work could stimulate new research directions in catalysis, nanomedicine, drug delivery and cancer therapy. To support the ambitious claims of this work, the authors present an impressive amount of experimental data. These results are well presented and described. Most of the claims are supported by the data presented. Moreover, the data are well discussed with respect to the previous literature. However, some important points in this work are not convincing for the reviewer. They should be reconsidered and clarified according to the comments below. Therefore, I recommend this manuscript for publication in Nature Communications after revision.

Major points

1. The main problem is related to the formulation of nano-Pd in form of PLGA nanoparticles. The authors showed in supplementary Fig. 2 that the release of Pd catalyst from NPs is slow and they concluded that the catalyst is well encapsulated. However, these measurements were done in PBS, where this catalysis is not well soluble. This can explain why the release was slow. In fact, the authors claim that "it is extremely lipophilic with a computed octanol-water partition coefficient (c

lop P) of 7.0 and very poor water solubility (0.02 mg/ml)", which confirms my concern. If Pd catalyst is very poorly soluble in water, it cannot be released to water even from a poorly stable formulation. Therefore, the experiments should be done in a model release media, that could solubilize the Pd-catalyst, for instance in the presence of surfactants, BSA, serum medium, alcohols or other type of solubilizing agents. Otherwise, the present data do not really provide sufficient proof for the stability of these nano-formulations in biological conditions. Moreover, for some reasons the authors do not clearly state that catalysis with Pd should take place only after the catalyst is released from the particle (or at least exposed at the NP surface) in order to interact with the substrate. This means that in any case, the positive results with nano-Pd would be possible only for particle able to release slowly the catalyst or to expose it at NP surface. This point should be discussed.

2. The role of nano-formulations for the catalysis in vivo is not clear, because the authors did not show the data on the effect of the Pd catalyst alone without polymer NPs. I could argue that iv injection of Pd catalyst in PBS could produce similar effects as for PLGA-PEG nano-formulations. In fact, this point is connected with point 1. If the nano-formulation is not stable in serum (or relevant model medium), the catalysis would be immediately released, so that PLGA-PEG NPs would play only marginal role in the biodistribution of the catalyst. Therefore, the authors should make corresponding control experiments and show that PLGA NPs are really needed to observe desired catalytic reactions in vivo. The reviewer understands that making these controls for all mice experiments would be just too complicated to realize. Therefore, the authors could limit the control experiments, for instance, to the fluorescent reporter Alloc2R110. If the nano-formulation in form of polymer NPs is really needed, the claims of this work would sound more solid. If the nano-formulation is not needed, then the claims related to nano-Pd should be revised and, even in this case, the work remains of the highest interest for Nature Communications.

Minor points

1. Figure 2b: only two particles are seen in the image. Why not to show larger image, where more particles could be visible? Two particles are not enough to judge the size and monodispersity of the obtained NPs. In the text it is stated that particles of 57 +/-2 nm were formed according TEM, mentioning that "n=3". What is "n": number of particles, images or experiments? In any case, it should be stated how many particles were analyzed. Moreover, the error appears really small, which is another reason to show a larger TEM image with many particles in order to judge their monodispersity.

2. Expression "several cell-lengths" is not clear.

3. In the results related to pro-drug activation, the authors state: "indicating the prodrug had not covalently reacted with cellular proteins". Then immediately after, they wrote: "DOX was free to covalently react with DNA". The problem is probably with the first sentence, where "proteins" should be replaced with "DNA".

4. Supplementary Fig. 4. Term "vehicle control" should be clarified.

5. Figure 3 is too large and will probably not be readable in the publication format. Its dimensions / organization should be optimized.

Thank you for the careful consideration of our manuscript NCOMMS-16-23593, “Nanopalladium is a cellular catalyst for in vivo chemistry,” which we had submitted to *Nature Communications*. The following is a point-by-point response to reviewer comments. We thank the reviewers and editors for their time, attention, and valuable input. We sincerely believe the manuscript is stronger after addressing their concerns and suggestions.

Reviewer #1 (Remarks to the Author):

The manuscript of Weissleder describes a novel concept, namely a nanoformulation of palladium that targets a tumor and subsequently catalyzes the conversion of a prodrug in a drug which leads to tumor regression. This is certainly an interesting approach and the authors have investigated in depth the performance of the formulation. The question is however whether this an approach potentially suitable for clinical translation. The following questions are raised and remarks are made.

1. The palladium catalyst was loaded in pegylated PLGA nanoparticles. It is know that from many studies that such particles accumulate in tumor by the so-called EPR effect. However at most 1 % of the injected dose (particularly in humans) end up in the tumor, This means that the particles are also deposited in other tissues and organs. The Pd-loaded nanoparticles will cause toxic side effects after administration of the prodrugs.

The reviewer is correct in describing the limitations of EPR, particularly in the clinic. The point of this manuscript is not to create more highly selective targeted nanoparticles that specifically accumulate in tumors, but rather to accomplish the more general goal of demonstrating *in vivo* palladium catalysis. Nonetheless, we note that our strategy uses a new dual stage NP treatment, which apparently increases tumor selectivity and selective pro-drug activation based on our data (Fig. 6; S6; S14; S15). These data distinguish our approach from strategies that rely solely on the EPR effect.

2. It is not clear to what extent this approach results in better therapeutic outcome as compared to nanoparticles loaded with a cytostatic drug. Figure 5c shows that a nanoparticle formulation of PLGA-PEG of and as well as free drug are as effective in inhibition of the tumor growth.

I therefore do not see therapeutic advantages (better efficacy and/or reduced toxicity compared to standard chemotherapies).

The reviewer is correct that efficacy is *as good as* standard chemotherapeutic formulations. However, there was significant beneficial effect on toxicity (Fig. 6a) — where myeloablation was completely avoided. This effect alone would be a considerable advantage in clinical practice and could also allow dose escalation.

A treatment of standard-formulated chemotherapy (30 mg/kg solvent-doxorubicin) is known to be quite above the maximum tolerated dose in mice. For instance, in a recent study, 24 mg/kg caused severe toxicity that prematurely ended the experiment (Fig. S14

of ref: [Xu et al., 2016, Nat Biotechnol, 34, 414-8]). Repeating such an experiment would cause unacceptable animal distress and is not permitted by our animal care review board (IACUC). In fact, we show that even 10 mg/kg dose causes myeloablation.

In contrast, the equivalent high dose (30 mg/kg) using the pro-drug Pd strategy was well tolerated.

Therefore, to address this point we have added the following text:

“Given the favorable safety profile of the Pd-NP and proDOX treatment, we increased proDOX dose to 300% the maximum tolerated dose (MTD) of un-encapsulated solvent-DOX in nu/nu mice, from 16 $\mu\text{mol/kg}$ (which equates to 10mg/kg DOX[Vanhoefer et al., 1996, Clin Cancer Res, 2, 1961-8]) to 48 $\mu\text{mol/kg}$. Of note, even the lower dose of 38.5 $\mu\text{mol/kg}$ liposomal doxorubicin nanoparticle (DOXIL) has been found to cause severe and prohibitive toxicity in 100% of animals treated in similar mouse models [Xu et al., 2016, Nat Biotechnol, 34, 414-8]. In contrast, Pd-NP combined with the 48 $\mu\text{mol/kg}$ dose of proDOX showed efficacy that was equivalent to treatment with either or nano-encapsulated or solvent-DOX (Fig. 5c). These results were further substantiated by tumor-weight measurements at the end of treatment (Fig. 5d; see Supplementary Fig. 13a for representative tumor photos).”

“Only un-encapsulated, solvent-DOX caused significant lymphocytopenia, while both solvent-DOX and DOX-NP led to total reticulocyte ablation, which indicates bone marrow toxicity (Fig. 6a). Likewise, previous reports have found indications of hematological toxicity with 20-38.5 $\mu\text{mol/kg}$ liposomal doxorubicin nanoparticle (DOXIL) in similar mouse models [Xu et al., 2016, Nat Biotechnol, 34, 414-8]. Our toxicity data is supported by HPLC analysis of excised tissue, showing that the prodrug activation strategy leads to <10% the levels of activated DOX in bone marrow compared to solvent-DOX or DOX-NP treatments (Supplementary Fig. 14a).”

Reviewer #2 (Remarks to the Author):

In the manuscript “Nano-palladium is a cellular catalyst for in vivo chemistry”, Miller et al. presented a novel strategy to in situ activate the prodrug using palladium catalyst based chemistry. This work is of interest and novel in nanomedicine. Overall, the authors did very careful studies in terms of chemistry and biology. However, I have several concerns. I would like to recommend major revision.

Comments:

1. What is the loading of catalyst in PLGA-PEG NPs and their stability?

The following data was added and expanded:

“The highly lipophilic $\text{PdCl}_2(\text{TFP})_2$ compound loaded into the PLGA core with high efficiency (70% encapsulation; 365 nmol Pd per mg of polymer), with robust stability

such that Pd-NP did not aggregate or significantly degrade over time (Supplementary Fig. 2d), and with controlled Pd release such that 50% of the compound was released from the NP by 20 h in an in vitro release assay (Supplementary Fig. 2e)."

2. It seems that buffers can significantly impact the activity of catalyst. The authors screened the catalyst using HBSS and MEM. Why do not use tissue homogenates?

We have added a test of catalyst activity in tissue homogenates, presented in Fig. S1g.

"Furthermore, the stability of its activity in complex physiological solutions including fetal bovine serum (FBS; Supplementary Fig. 1f), and its activity in whole tumor homogenate (Supplementary Fig. 1g) could be improved. This may in part be explained by the observation that metal complexes such as PdCl₂(TFP)₂ undergo dynamic ligand exchange processes, and ligands including TFP and Cl may be replaced by biological components (Supplementary Fig. 1c). Thus, we aimed to improve catalyst delivery while restricting its interaction with potentially reactive biological material in a clinically applicable manner. We encapsulated the pre-catalyst in a polymeric nanoformulation based on materials that have entered clinical trials [Hrkach et al., 2012, Sci Transl Med, 4, 128ra39] and using polymers that had previously been approved by the FDA for use in clinical NP preparations: poly(lactic-co-glycolic acid) (PLGA) and PLGA-PEG."

In the context of this statement, we note that nano-encapsulation improves Pd compound delivery (Supplementary Fig. 3a) and stability (Supplementary Fig. 3b); improves in vivo activity in tumors (Supplementary Fig. 8e); and leads to ~50% prodrug activation both in tissue-culture experiments (Supplementary Fig. 12) and in xenograft models (Supplementary Fig. 13d). Thus, activity is retained in complex biological buffers.

3. It is very surprising that the activation of prodrug only occurred in tumor while not in other organs, such as liver and spleen. The mechanism must be clarified clearly.

To clarify the mechanism, we performed high magnification confocal fluorescence microscopy of catalyst accumulation and activity in the liver and kidney. We find that although the Pd-NP and substrate indeed accumulate in these organs, reflecting what is observed with similar nanoformulations, in the liver this uptake is primarily confined to resident phagocyte populations (e.g., Kupffer cells) rather than hepatocytes (Supplementary Fig. 15). Furthermore, imaging suggests that the catalyst and its substrates may accumulate in different cellular or subcellular compartments in both the kidney and the liver (Supplementary Fig. 15), which could potentially be the result of active drug pumps highly expressed in those tissues. This is in contrast to what is observed in tumor cells, where both the catalyst and substrate fluorescence show substantial spatial overlap (Supplementary Fig. 8-10).

"We imaged catalyst accumulation and activity in the liver and kidney to better understand why nuclear drug accumulation was not observed in these organs. We found that although Pd-NP and its substrate indeed accumulate, in the liver this uptake

is primarily confined to resident phagocyte populations rather than hepatocytes (Supplementary Fig. 15). Furthermore, imaging suggests that the catalyst and its substrates may accumulate in different cellular or subcellular compartments in both the kidney and the liver (Supplementary Fig. 15), which could potentially be the result of active drug pumps highly expressed in those tissues. This is in contrast to what is observed in tumor cells, where both the catalyst and substrate fluorescence show substantial spatial overlap (Supplementary Fig. 8-10)."

4. Why were NPs administered into OVCA tumor intraperitoneally and subcutaneous human fibrosarcoma tumors intravenously? The authors should discuss why NPs were administered differently.

Recent clinical research has highlighted the potential for intraperitoneal (rather than or in addition to intravenous) chemotherapy to improve outcomes in OVCA patients. Although this strategy carries additional risks of complication, it has been repeatedly associated with improvement in overall survival [Wright et al., 2015, J Clin Oncol, 33, 2841-7]. We hypothesize that this effect may be even more substantial for nanomedicines, and therefore used i.p. injection to treat OVCA. The text has been updated accordingly:

"Tissue was analyzed 24 h after Pd-NP administration in two xenograft mouse models. Pd-NP was intravenously (i.v.) injected via the tail vein in mice bearing subcutaneous human fibrosarcoma tumors. In ovarian cancer (OVCA) patients, recent clinical research has highlighted the potential for intraperitoneal (i.p.) chemotherapy to improve outcomes [Wright et al., 2015, J Clin Oncol, 33, 2841-7]. We therefore administered Pd-NP i.p. in an orthotopic model of disseminated OVCA using the human ES2 cell line (Supplementary Fig. 5d)."

Reviewer #3 (Remarks to the Author):

The manuscript reports on the catalytic prodrug activation that is triggered by a palladium complex, which was incorporated in nanoparticles. On the one hand the general impression is that the work shows significant problems regarding chemistry and structural characterization. On the other hand the manuscript reports on a very impressive biological study, which sheds light on the biological properties of palladium containing nanoparticles and their catalytic properties in-vitro and in-vivo. Despite the fascinating biological results, the work does not stand on a very solid basis as the most important compound(1) was not characterized in an appropriate manner. Therefore, recommendation for publication can not be given.

The critical remarks in more detail:

- 1) The purity of the palladium complexes was not determined. This a "no go" regarding medicinal chemistry and drug discovery. Even small impurities can influence the outcome of biological tests significantly. Especially when dealing with probably non stable coordination compounds, high purity of all test compounds is critical.

We thank the reviewer for pointing out our failure to include purity measurements. We indeed routinely measure purities for all synthesized compounds. We have now added the following text to describe purity of all tested catalysts, and synthesis references are listed for all catalysts:

“All catalysts were characterized by NMR for purity and successful synthesis following previously described protocols. In all cases purity was found to be >99%. For the most important compound 1, prepared as described by Hettrick and Scott [Hettrick and Scott, 1991, J Am Chem Soc, 113, 4903-4910], NMR, ESI-MS, and elemental analysis are included below. Stability and purity of all catalysts were monitored by NMR throughout the entire study to avoid and eliminate any effects caused by degradation products or impurities.”

- 2) The structural characterization of the new compounds lacks sufficient evidence.
- a) For example, in the ^1H NMR data of compound 1 several signals with an intensity of 0.3 were observed. What is the origin of these signals ?

These signals were due to the cis/trans mixture of the catalyst, and agree with the previously described data of this compound that includes a crystal structure [Hettrick and Scott, 1991, J Am Chem Soc, 113, 4903-4910]. This crystal structure depicted cis stereochemistry about a square planar Pd conformation; NMR spectra had previously indicated a mixture of cis/trans isomers, with majority in the cis conformation [Hettrick and Scott, 1991, J Am Chem Soc, 113, 4903-4910].

The adequate and accurate characterization reported in Hettrick and Scott has been cited in subsequent review articles [Andersen and Keay, 2001, Chem Rev, 101, 997-1030] on TFP catalysis as *the* key reference, stating “Pd(TFP)₂Cl₂ has been isolated and characterized (Hettrick and Scott, 1991).”

The text has been updated as follows:

*“NMR (DMSO-*d*₆, r.t., [ppm]) ^1H NMR (400 MHz): δ = 8.14-8.05 (s (br), 0.3 H), 7.97-7.89 (m, 3H), 7.24-7.12 (s (br), 0.3 H), 6.98-6.88 (m, 3H), 6.74-6.64 (s (br), 0.3 H), 6.62-6.54 (m, 3H). ^{13}C NMR (101 MHz): δ = 149.9 (d, JP-C = 5.3 Hz), 139.9 (d, JP-C = 92.4 Hz), 124.8 (d, JP-C = 20.1 Hz), 111.7 (d, JP-C = 7.8 Hz). Intensity 0.3 H and 3H peaks correspond to trans and cis isomers, respectively, in agreement with ref. [Hettrick and Scott, 1991, J Am Chem Soc, 113, 4903-4910]. ^{31}P NMR (162 MHz): δ = -22.0 (cis, >90%, ref. [Hettrick and Scott, 1991, J Am Chem Soc, 113, 4903-4910]), -28.9 (trans, <10%).”*

- b) There are no mass spectrometry data and there is no experimental evidence for the presence of the chlorine ligands of 1.

The synthesis procedure was followed as described previously [Hettrick and Scott, 1991, J Am Chem Soc, 113, 4903-4910], with consistent ^1H and ^{31}P NMR as in that

report. We have added mass spectrometry and elemental analysis, all in agreement with the previous publication:

“Anal. Calcd for $C_{24}H_{18}Cl_2O_6P_2Pd$: C, 44.92; H, 2.83. Found #1: C, 45.15; H, 2.95. Found #2: C, 45.03; H, 2.88 (average results differ from calculated by <2%). LC-MS (ESI) calc for $C_{24}H_{18}Cl_1O_6P_2Pd [M-Cl]^+$ 604.93, found 605.0 (in agreement with ref. [Hettrick and Scott, 1991, J Am Chem Soc, 113, 4903-4910]). Of note, predominant cis-stereochemistry about a square planar Pd conformation was previously confirmed by crystal structure [Hettrick and Scott, 1991, J Am Chem Soc, 113, 4903-4910].”

c) The aspect of cis/trans isomerism was not evaluated appropriately. Figure 2a shows a trans-configured synthesis precursor and a cis-configured complex PdCl₂(TFP)₂. P-NMR data of 1 suggest a mixture of cis- and trans-isomers with the trans-isomer as impurity.

We thank the reviewer for this oversight, and have corrected the Figure 2a. Indeed a cis/trans mixture was used.

d) The structure of compound proDOX was only checked by NMR spectroscopy, which is insufficient as a stand-alone characterization technique.

We initially excluded other data because the compound's synthesis and characterization are established [Cotterill et al., 2008, Biotechnol Bioeng, 101, 435-40]; nonetheless, mass spectrometry had been performed as a routine matter, showing appropriate results, which are now included.

“LC-MS (ESI) calc for $C_{31}H_{32}NO_{13} [M-H]^-$ 626.2, found 626.06 (Supplementary Fig. 11a).”

3) The aspect of stability and metabolism under biological conditions was not touched at all. Metal complexes can undergo ligand exchange processes in biological systems. The Cl-ligands of 1 might be easily replaced by components of the cells or tissues resulting in different Pd-containing species.

We have expanded the analysis of catalyst stability to now include results in tissue homogenates (which was also suggested by reviewer 2). This reviewer is correct in suspecting the dynamic nature of the ligands — we now describe the dynamic nature in Fig. S1 describing the mechanism of catalysis, which has been defined in previous literature, and which describes replacement of the Cl ligands (see figure above). Components of cells or tissues may interject themselves into this process, and we have now stated this in the text. It is difficult to directly observe formation of such species in complex biological solutions, however we are able to monitor catalytic activity in complex mixtures as a metric of catalyst stability (Fig. S1f-g, S3b), and we monitor the formation of a major degradative product, tri(2-furyl)phosphine oxide, in physiological solutions (Fig. S1e).

“Furthermore, the stability of its activity in complex physiological solutions including fetal bovine serum (FBS; Supplementary Fig. 1f), and its activity in whole tumor homogenate (Supplementary Fig. 1g) could be improved. This may in part be explained by the observation that metal complexes such as PdCl₂(TFP)₂ undergo dynamic ligand exchange processes, and ligands including TFP and Cl may be replaced by biological components (Supplementary Fig. 1c). Thus, we aimed to improve catalyst delivery while restricting its interaction with potentially reactive biological material in a clinically applicable manner. We encapsulated the pre-catalyst in a polymeric nanoformulation based on materials that have entered clinical trials [Hrkach et al., 2012, Sci Transl Med, 4, 128ra39] and using polymers that had previously been approved by the FDA for use in clinical NP preparations: poly(lactic-co-glycolic acid) (PLGA) and PLGA-PEG.”

In the context of this statement, we note that nano-encapsulation improves Pd compound delivery (Supplementary Fig. 3a) and stability (Supplementary Fig. 3b); improves in vivo activity in tumors (Supplementary Fig. 8e); and leads to ~50% prodrug activation both in tissue-culture experiments (Supplementary Fig. 12) and in xenograft models (Supplementary Fig. 13d). Thus, activity is retained in complex biological buffers.

4) In the discussion section a 5-20 micromolar intratumoral Pd-NP concentration is mentioned. How was this calculated ?

This calculation was determined by converting known %ID/g measurements (ICP-MS, and fluorescence in the case of the similarly-sized NP vehicle) into concentrations by assuming a standard tissue density. We have expanded the text accordingly.

“Based on ICP-MS observation of 0.3-1.25% ID/g in tumor cells of multiple xenograft models (Supplementary Fig. 5d), we found an intratumoral Pd-NP concentration of 5-20 μM at the MTD of 50 mg/kg (calculated from %ID/g and assuming standard tissue density), which is well above the concentrations at which we observed efficient catalytic activity, yet no organ accumulated Pd-NP at levels observed to be toxic in vitro, and no toxicity was observed in treated animals.”

*In Methods: "Molar concentrations in tissue assumed density of 1 kg / L and were estimated as [%I.D./g] * [total mass I.D.] * density_{tissue} / MW."*

Reviewer #4 (Remarks to the Author):

The manuscript by Miller et al describes palladium (Pd) complex formulated with polymer nanoparticles (NPs) as catalyst for chemical reactions in vivo. The authors claim that the obtained nano-palladium is able to catalyze activation of fluorescent dyes and liberation of active drug at the cellular and animal levels. It was also shown that nano-palladium in combination with doxorubicin prodrug can significantly improve specific targeting of tumors in mice: the active drug was liberated specifically inside tumors producing inhibition of the tumor growth with minimized toxicity in other organs, in contrast to conventional formulations of this drug. The work presents several novel claims. It was shown that nano-palladium can operate in physiological conditions and even more importantly, it can catalyze reactions in mice, which is probably the key novelty of this work. Moreover, the important novel observation is the remarkable specificity of tumor targeting with minimal toxicity in other tissues, which is connected to the drug activation through Pd-catalysis. This work will be definitely of interest for a broad community of researchers. It will be of interest for chemists working on metal-catalysis and those working on bio-orthogonal reactions. It will be also of interest for biologists, nanotechnologists, pharmacologists and biomedical researchers, developing new tools against cancer, etc. The work could stimulate new research directions in catalysis, nanomedicine, drug delivery and cancer therapy. To support the ambitious claims of this work, the authors present an impressive amount of experimental data. These results are well presented and described. Most of the claims are supported by the data presented. Moreover, the data are well discussed with respect to the previous literature. However, some important points in this work are not convincing for the reviewer. They should be reconsidered and clarified according to the comments below. Therefore, I recommend this manuscript for publication in Nature Communications after revision.

We thank the reviewer for their positive comments and helpful suggestions.

Major points

1. The main problem is related to the formulation of nano-Pd in form of PLGA nanoparticles. The authors showed in supplementary Fig. 2 that the release of Pd catalyst from NPs is slow and they concluded that the catalyst is well encapsulated. However, these measurements were done in PBS, where this catalysis is not well soluble. This can explain why the release was slow. In fact, the authors claim that "it is extremely lipophilic with a computed octanol-water partition coefficient (c_{lop} P) of 7.0 and very poor water solubility (0.02 mg/ml)", which confirms my concern. If Pd catalyst is very poorly soluble in water, it cannot be released to water even from a poorly stable formulation. Therefore, the experiments should be done in a model release media, that could solubilize the Pd-catalyst, for instance in the presence of surfactants, BSA, serum medium, alcohols or other type of solubilizing agents. Otherwise, the present data do

not really provide sufficient proof for the stability of these nano-formulations in biological conditions.

To address this issue, we measured catalyst release across a panel of relevant solutions, including 10% BSA, full growth media (DMEM + 10% FBS), and solubilizing agents such as DMF, ethanol, and triton-X, and in acidic conditions such as are found in endosomes:

“We also measured Pd compound release across a panel of relevant solvents, solubilizing agents, and biologically-relevant buffer conditions. Most factors had little effect, on average changing release rates by <10% (Supplementary Fig. 2f), suggesting the particles were stable under a variety of environments. BSA increased the release rate by 65%, likely serving as a solubilizing agent, but this effect was still modest compared to the impact of ethanol and high levels of particle-dissolving dimethylformamide (DMF) (Supplementary Fig. 2f).”

Moreover, for some reasons the authors do not clearly state that catalysis with Pd should take place only after the catalyst is released from the particle (or at least exposed at the NP surface) in order to interact with the substrate. This means that in any case, the positive results with nano-Pd would be possible only for particle able to release slowly the catalyst or to expose it at NP surface. This point should be discussed.

Thank you for the excellent point which we now expand on in the discussion. Indeed, it is a key component of the “localized” nature of the strategy. In the context of discussing Pd compound release from the NP, we have added the following: *“This evidence for NP stability and controlled release is important for successful Pd activity, since catalysis cannot occur if both the Pd compound and its substrate remain separately encapsulated in their distinct NP vehicles.”*

This point is re-iterated in text describing the *in vivo* time-lapse imaging of activity in tumors: *“Either the Pd pre-catalyst or the Alloc₂R110 need to be released from their respective polymer NP vehicles in order for the catalytic reaction to occur; otherwise the reactant and pre-catalyst remain isolated in their spatially separated nano-encapsulations.”*

2. The role of nano-formulations for the catalysis *in vivo* is not clear, because the authors did not show the data on the effect of the Pd catalyst alone without polymer NPs. I could argue that *iv* injection of Pd catalyst in PBS could produce similar effects as for PLGA-PEG nano-formulations. In fact, this point is connected with point 1. If the nano-formulation is not stable in serum (or relevant model medium), the catalysis would be immediately released, so that PLGA-PEG NPs would play only marginal role in the biodistribution of the catalyst. Therefore, the authors should make corresponding control experiments and show that PLGA NPs are really needed to observe desired catalytic reactions *in vivo*. The reviewer understands that making these controls for all mice experiments would be just too complicated to realize. Therefore, the authors could limit the control experiments, for instance, to the fluorescent reporter Alloc₂R110. If the

nano-formulation in form of polymer NPs is really needed, the claims of this work would sound more solid. If the nano-formulation is not needed, then the claims related to nano-Pd should be revised and, even in this case, the work remains of the highest interest for Nature Communications.

We thank the reviewer for this suggestion, and have performed additional control experiments using un-encapsulated palladium complex *in vivo*. Results show that Pd-mediated fluorogenic Alloc₂R110 uncaging in the tumor is decreased by 90% when the Pd compound is administered in an un-encapsulated form. Similar key feasibility experiments were also done early during this research and led us to the encapsulation strategy in the first place. From the text, “*To further test the influence of nano-encapsulation on catalytic activity, we examined whether an i.v. infusion of an un-encapsulated, solvent-based Pd compound formulation could similarly uncage Alloc₂R110 in tumors. Results show the solvent-based administration was 85% less efficient (Supplementary Fig. 8e).*”

Minor points

1. Figure 2b: only two particles are seen in the image. Why not to show larger image, where more particles could be visible? Two particles are not enough to judge the size and monodispersity of the obtained NPs. In the text it is stated that particles of 57 +/-2 nm were formed according TEM, mentioning that “n=3”. What is “n”: number of particles, images or experiments? In any case, it should be stated how many particles were analyzed. Moreover, the error appears really small, which is another reason to show a larger TEM image with many particles in order to judge their monodispersity.

We have added additional TEM images showing more particles, and quantified the size distribution accordingly.

“*Transmission electron microscopy (TEM) enabled direct visualization of Pd-NP (Fig. 2b; see Supplementary Fig. S2b-c for more images and distribution). By TEM, Pd-NP appear smaller after staining (27 ± 7 nm mean \pm std. dev., $n=73$), which is consistent with similar ~50% size decreases that have been documented for other polymeric NP [Miller et al., 2015, Nat Commun, 6, 8692].*”

2. Expression “several cell-lengths” is not clear.

We have added a supplemental figure to demonstrate the calculation and the length scale we describe.

“*Despite only a minority of Alloc₂R110 NP extravasating into the tumor tissue at early time points ($t < 2$ h), the majority ($68\% \pm 4\%$, mean \pm SEM across $n = 3$) of fluorescence turn-on actually occurred within the tissue itself, and highest turn-on was observed in*

tumor tissue near (<100 μm) functionally perfused microvasculature (Supplementary Fig. 9)."

3. In the results related to pro-drug activation, the authors state: "indicating the prodrug had not covalently reacted with cellular proteins". Then immediately after, they wrote: "DOX was free to covalently react with DNA". The problem is probably with the first sentence, where "proteins" should be replaced with "DNA".

We have changed the text accordingly.

4. Supplementary Fig. 4. Term "vehicle control" should be clarified.

We have expanded the text accordingly. Vehicle control refers to PLGA-PEG NP injection.

5. Figure 3 is too large and will probably not be readable in the publication format. Its dimensions / organization should be optimized.

We have re-arranged the figure.

REVIEWERS' COMMENTS:

Reviewer #2 (Remarks to the Author):

I am satisfied with the revision and recommend it for publication.

Reviewer #3 (Remarks to the Author):

The authors have fully addressed all my previous critical concerns in a very professional manner. At the end, the catalyst was applied as a 9/1 mixture of cis/trans isomers. I think this should be mentioned appropriately in the main text of the work in order to avoid misinterpretations. However, this is a decision of the authors. I have no further remarks and look forward to read the published paper.

Reviewer #4 (Remarks to the Author):

In the revised manuscript, the authors took into account all my comments. Notably, they made requested additional experiments and the obtained new results supported well their claims. Therefore, I recommend this manuscript for publication in Nature Communications in the present form.

Dear editors,

Please find below the point-by-point response to the reviewer comments.

Thank you again for your time and attention,

Ralph Weissleder

REVIEWERS' COMMENTS:

Reviewer #2 (Remarks to the Author):

I am satisfied with the revision and recommend it for publication.

Reviewer #3 (Remarks to the Author):

The authors have fully addressed all my previous critical concerns in a very professional manner. At the end, the catalyst was applied as a 9/1 mixture of cis/trans isomers. I think this should be mentioned appropriately in the main text of the work in order to avoid misinterpretations. However, this is a decision of the authors. I have no further remarks and look forward to read the published paper.

We have added this to the text:

“Although cis-isomer has been isolated and crystallized using the same synthesis method used in this manuscript¹⁸, **and a 9:1 cis/trans isomer mixture was used herein (see *Methods*)**, many applications of TFP as a coordinating ligand utilize free TFP for *in situ* catalyst formation¹⁶.”

Reviewer #4 (Remarks to the Author):

In the revised manuscript, the authors took into account all my comments. Notably, they made requested additional experiments and the obtained new results supported well their claims. Therefore, I recommend this manuscript for publication in Nature Communications in the present form.